# Modelling Global Tropical Cyclone Wind Footprints

James M. Done[1,2], Ming Ge[1], Greg J. Holland[1,2], Ioana Dima-West[3], Samuel Phibbs[3], Geoffrey R. Saville[3], and Yuqing Wang[4]

[1]National Center for Atmospheric Research, 3090 Center Green Drive, Boulder CO 80301, US
[2]Willis Research Network, 51 Lime St, London, EC3M 7DQ, UK
[3]Willis Towers Watson, 51 Lime St, London, EC3M 7DQ, UK
[4]International Pacific Research Center and Department of Atmospheric Sciences, School of Ocean and Earth Science and Technology, University of Hawaii at Manoa, HI 96822, US.

*Correspondence to*: James M. Done (done@ucar.edu)

**Abstract**. A novel approach to modelling the surface wind field of landfalling tropical cyclones (TCs) is presented. The modelling system simulates the evolution of the low-level wind fields of landfalling TCs, accounting for terrain effects. A two-step process models the gradient-level wind field using a parametric wind field model fitted to TC track data, then brings the winds down to the surface using a numerical boundary layer model. The physical wind response to variable surface drag and terrain height produces substantial local modifications to the smooth wind field provided by the parametric wind profile model. For a set of U.S. historical landfalling TCs the accuracy of the simulated footprints compare favourably with contemporary modelling approaches. The model is applicable from single event simulation to the generation of global catalogues. One application demonstrated here is the creation of a dataset of 714 global historical TC overland wind footprints. A preliminary analysis of this dataset shows regional variability in the inland wind speed decay rates and evidence of a strong influence of regional orography. This dataset can be used to advance our understanding of overland wind risk in regions of complex terrain and support wind risk assessments in regions of sparse historical data.

## 1 Introduction

Tropical Cyclones (TCs) dominate U.S. weather and climate losses (Pielke Jr. et al., 2008; Smith and Katz, 2013). They account for 41 % of the inflation-adjusted U.S. insured loss between 1995 and 2014. Future increases in TC peak wind speeds (Walsh et al., 2016), in combination with rapid population increases, mean TC wind losses are set to rise even further (Geiger et al., 2016; Estrada et al., 2015; Ranson et al., 2014; Weinkle et al., 2012). Improved approaches to assessing overland TC wind fields is needed to enable society to manage this increasing risk.

While coastal communities may experience relatively frequent TC impacts, inland communities experience TC impacts far less often and less is known about the likelihood of inland damaging winds. Given the scientific consensus that average TC wind speeds will increase in the future (e.g., Villarini and Vecchi, 2013; Murakami et al., 2012; Hill and Lackmann, 2011; Elsner et al., 2008) and that category four and five hurricanes have increased substantially in recent decades (Holland and Bruyère, 2014), strong winds may be experienced farther inland in the future, all other TC and environment characteristics being equal. Modelling approaches that capture TC footprints - the entire overland swath of storm-lifetime maximum wind speed from the immediate coast to far inland - is therefore a key need.

New views of global TC footprints are critically needed to support a variety of risk management activities. One particular need is to characterize overland footprints for mountainous countries that have both a high TC risk and significant insurance exposure, such as the Philippines and Japan. What is the impact of coastal terrain features on TC wind distributions and potential losses? And how does terrain affect overland extreme wind probabilities? In addition, a catalogue of global historical events may also be used to model losses from historical events. These scenarios stress-test reinsurance structures to ensure companies have adequate protection, and can also be used in submissions to regulators. Long records of TC overland wind

footprints also inform the generation of synthetic event sets (particularly in regions of sparse historical data), and inform near- and long-term views of wind probability accounting for climate variability and incorporating the effects of climate change. A global catalogue of TC wind footprints is also needed to advance our basic understanding of TC climate across basins. For example, what are the global- to local-scale processes controlling regional spatial and temporal trends and variability of overland TC winds?

Using an analytical boundary layer model to simulate the low-level winds during Hurricane Fabian (2003) over Bermuda, Miller et al. (2013) found winds at the crest of a ridgeline at category four strength compared to category two strength in simulations without terrain. Simulations of Cyclone Larry (2006) over the coastal ranges of Queensland, Australia using a full numerical weather prediction model by Ramsay and Leslie (2008) also produced wind speed-ups along hill crests and windward slopes. The high Froude number flow brought about by the high wind speeds and quasi-neutral stability causes flow directly over the terrain features with minimal lateral displacement. Under mass continuity, flow accelerates as the air column thins passing over higher terrain. This speed-up also supports wind-shear driven turbulence and enhances peak gusts. For high mountains, however, the wind has a greater potential to become blocked. While a neutral boundary layer is not guaranteed at large radii (e.g., Kepert (2012) indicates increasing static stability as subsidence increases at large radii) measurements of turbulent fluxes in high-wind environments between outer rain bands by Zhang et al. (2009) find shear production and dissipation to be the dominant source and sink terms of TKE.

Given that the work done by the wind in directly damaging structures varies by the cube of the wind speed (Emanuel, 2005), terrain effects on damage have the potential to be significant. Indeed, Miller et al. (2013) found that the greatest residential roof damage was located along the ridgeline and the windward slopes of Bermuda. Terrain effects were also found in residential wind damage patterns during Cyclone Larry in 2006 (Henderson et al., 2006) and during Hurricane Marilyn in 1995 across the Island of St Thomas in the Caribbean (Powell and Houston, 1998). Incorporating terrain effects could therefore improve wind risk assessments when compared to traditional analytic methods in regions of complex terrain, thus supporting potential losses assessments and underwriting decisions in re/insurance markets.

Current practice in wind field modelling spans a range of complexity, depending on the application. The simplest models, known as parametric radial wind profiles fit functions to a small number of readily available TC and environmental parameters to characterize the radial profile of wind and pressure from the TC centre. The Holland et al. (2010) profile, for example, models the surface winds directly whereas the Willoughby et al. (2006) profile models the gradient-level winds and an extra step is needed to determine the surface winds. These models are computationally efficient and therefore widely used as the hazard component of catastrophe models (Mitchell-Wallace et al., 2017; Vickery et al., 2009), and can be used to compute wind exceedance probabilities anywhere on Earth. But fast computation comes at a price. The resulting wind fields are smooth, and so empirical corrections are typically applied to represent surface terrain effects.

An alternative approach is a reanalysis of observations. A reanalysis is created using a physical model that is nudged towards available observations. While a reanalysis produces gridded data and may capture observed asymmetries, it may still miss the effects of a variable surface roughness [e.g., HWIND (Powell et al., 1998) is representative only of open terrain], and to date only a small fraction of historical global events has been reanalysed. Another alternative approach is geostatistical spatial modelling. This data-driven approach combines spatial statistics to capture spatial dependence with extreme value theory to capture peak wind speeds. Again, this model is highly efficient but so far has only been developed for European windstorms (Youngman and Stephenson, 2016) to the authors' knowledge. Finally, four-dimensional high-resolution numerical modelling captures many more physical processes (e.g., Davis et al., 2010). But it is computationally too expensive and can be used only in a small number of cases. It is therefore of marginal use as the hazard component of catastrophe models, aside from use to develop improved parametric models (Loridan et al., 2015; Loridan et al., 2017).

This paper describes a novel and globally applicable approach to modelling the surface wind field of landfalling TCs. The model was developed as a collaboration between atmospheric scientists and reinsurance industry experts to ensure the model and resulting datasets are readily applicable to decision-making processes and based in peer-reviewed science. The modelling system combines the high efficiency of the parametric profile model with a representation of the interaction of the flow and variable surface terrain that captures the essential dynamics and physics. The modelling system simulates the temporal evolution of the near-surface spatial wind fields of landfalling TCs, accounting for terrain effects such as coastal hills and abrupt changes in surface roughness due to coastlines, forested or urban areas. The approach fits a parametric wind field model to historical or synthetic TC track data, and captures the frictional response of the wind field to the Earth's surface using a three-dimensional numerical model of the lowest 2-3 km of the atmosphere.

Application of the model is demonstrated through the creation of a dataset of 714 historical landfalling TC footprints globally. Such global footprint datasets have been created before but none used a non-linear boundary layer model that captures the dynamical response to a variable lower boundary. Giuliani and Peduzzi (2011) utilized a dataset of global historical TC footprints generated using a parametric model. More recently, Tan and Fang (2018) generated a dataset of 5376 global historical footprints using an approach that simulates the gradient winds using a parametric wind profile model and bringing the winds down to the surface using a simple power law profile that depends on the local surface roughness (Meng et al., 1997). Terrain effects were included using a simple speed-up factor based on four categories of terrain type and wind direction. By using a three-dimensional model our approach includes additional physical terrain effects. Our approach only considers the lowest 2-km of the atmosphere, thereby excluding the free troposphere needed for large-scale mountain waves to bring free-atmosphere winds down to the surface. It's also unlikely that our modelling approach has the resolution to capture flow-separation turbulence downwind of crests and escarpments. However, the physical terrain effects permitted by the model include convergence, vertical diffusion and vertical advection on windward slopes and crests resulting in locally strong low-

level shear and TKE production. In addition, the vertical boundary layer structure allows the potential for super-gradient jets (Franklin et al. 2003; Kepert and Wang 2001) to influence winds in high terrain. Finally, the time dimension allows for upwind effects due to upwind terrain variations and terrain to be incorporated.

The next section describes the modelling approach. Section 3 presents sensitivity simulations for a case study of Hurricane Maria (2017) over Puerto Rico to demonstrate the effects of adding the boundary layer model and variable terrain. A model evaluation against surface station observations is provided in Sect. 4. Section 5 describes the dataset of global historical landfalling TC footprints and includes a preliminary analysis. Finally, conclusions are presented in Sect. 6.

## 2 Method

A TC footprint is generated using a two-stage modelling process bookended by pre- and post-processing steps, as summarized by the flow diagram in Fig. 1. Stage one fits a parametric model of upper winds and pressure to the input TC track data. Stage two applies a three-dimensional numerical boundary-layer model to generate a detailed surface wind field incorporating the effects of terrain features such as coastlines, inland orography, and variable land surface friction.

The pre-processing step removes an estimate of the asymmetry due to storm motion ($V_a$) from the maximum wind speed input from the TC track data. The extent to which asymmetry due to forward speed is included in best track wind speed is uncertain. But given that these winds are Earth-relative measurements we assume that removing an uncertain estimate of asymmetry produces a more accurate estimate of the rotational wind than the original best track wind speed. The portion removed is a function of the TC translation speed ($V_t$), $V_a = 1.173 V_t^{0.63}$, following Chavas et al. (2017). The post-processing
step then adds back an estimate of the asymmetry due to storm motion to the output surface wind velocity field, again following Chavas et al. (2017). In addition, the fraction of this storm motion vector added is equal to one at the radius of maximum winds and then decays with increasing radius, following Jakobsen and Madsen (2004). Our approach therefore misses any interaction effects between terrain and the asymmetrical component of the storm wind field. The importance of these effects is unknown and we leave their inclusion for a future iteration of our model.

The final footprint is a map of the storm lifetime maximum one-minute average wind at 10 meters above the Earth's surface. The boundary layer model of Kepert and Wang (2001, hereafter KW01) outputs the instantaneous wind speed at 10 meters above the surface which is the lowest model level. While the instantaneous wind field output from a numerical model does not directly correspond to a specific averaging interval, some guidance is provided by the model timestep. A typical KW01
timestep of four seconds adequately resolves variability at timescales of a minute. The footprint is then simply calculated as the storm lifetime maximum wind speed at each grid point. Frequent model output intervals or a weak smoother may be needed to minimize the appearance of rings of strong winds in the footprint, particularly for fast moving TCs.

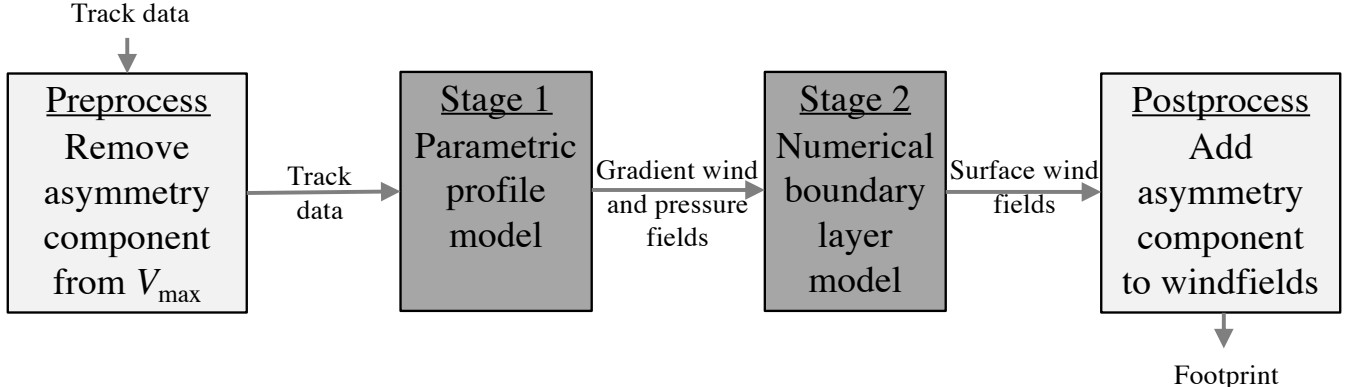

**Figure 1: Workflow diagram of the two-stage modelling process, bookended by pre- and postprocessing steps.**

## 2.1 Input Tropical Cyclone Track Data

The input TC track may be historical, synthetic or a real-time forecast. But the accuracy of the historical or forecast cyclone-scale footprints will be sensitive to the accuracy of the input track data. Typical historical data sources include the global International Best Track for Climate Stewardship (IBTrACS, Knapp et al., 2010), the Extended Best Track Dataset (Demuth et al., 2006, for the East Pacific and North Atlantic), or from the Joint Typhoon Warning Center (JTWC). Track data require latitude, longitude, maximum wind speed ($V_{max}$), radius of maximum wind ($R_{max}$), and environmental pressure. If the Holland et al. (2010) wind profile is used (discussed in the next section), an additional variable of the radius of 34 kt winds is required. Sensitivity tests (not shown) found that the model requires new track data every 10 minutes, to smooth out changes in the forcing of KW01 and reduce shocks. These can be obtained by simple interpolation.

Applying the modelling approach to create a dataset of global historical landfalling TC footprints requires as input the globally consistent IBTrACS v04 TC track dataset (Knapp et al., 2010). While Tan and Fang (2018) fill in missing variables using empirical relationships between TC variables, we choose to exclude tracks with missing data. For all basins, landfalling track points are identified using the landfall flag in the IBTrACS dataset (Knapp et al., 2010). Bypassing storms are included to capture the storms that don't make landfall, but still bring strong winds onshore. Such storms are identified using the 'distance to land' variable in IBTrACS, and defined as TCs that track within 50 km of a coastline, or within 250 km of the coastline with maximum wind speeds greater or equal to 50 kts (58 mph). Since our interest is in winds over land, storms are simulated

from approximately 12 hours before landfall (or before the closest point to land for bypassing storm) as far inland as the end of the track, or until the TC tracks back out over open water.

## 2.2 Stage 1: Modelling the Gradient-Level Wind and Pressure Fields

Initial solutions for the gradient-level spatial wind and pressure fields are created for each time step using a parametric profile model. The gradient-level solution represents upper winds unaffected by frictional and terrain effects from the lower surface boundary. One definition for the boundary layer height is the depth of the inflow, defined as the height where the radial inflow falls to 10 % of the peak inflow. Using radiosonde ascents in 13 hurricanes Zhang et al. (2011) find this height

to be approximately 850 m at the radius of maximum wind rising to approximately 1300 m at larger radii. Parametric profile models use functional radial profiles to determine the wind speed and pressure field from the TC centre [see Vickery et al., (2009) for an overview]. Our modelling approach is flexibly adaptable to use most choices of radial profile model. Sensitivity tests (not shown) with the Holland et al. (2010) and Willoughby et al. (2006) profiles showed some differences. The Holland et al. (2010) profile has the advantage of tying down the radial decay profile using an observation of an outer

wind, say the radius of 34 knot winds. However, observations of outer winds are not readily available globally. Willoughby et al. (2006) uses a sectionally continuous wind-profile comprising a power law inside the eye and two exponential decay functions outside. A polynomial smooths the transition across the radius of maximum wind.  This allows greater flexibility for using those databases without 34 kt wind radii. The viability of forcing KW01 with the Willoughby profile was demonstrated by Ramsay et al. (2009) for a case study simulation of Tropical Cyclone Larry (2006), by Kepert (2006a) for

Hurricane Georges (1998), Kepert (2006b) for Hurricane Mitch (1998), and Schwendike and Kepert (2008) for Hurricanes Danielle (1998) and Isabel (2003). Willoughby et al. (2006) conducted a comprehensive evaluation using flight level data. The verified performance of Willoughby motivated our choice of Willoughby profile for all simulations presented in this paper.

The outer radius of damaging winds can be highly sensitive to the choice of free parameters. The length scale for the transition region across the eyewall is set to 25km when $R_{max}$ is greater than 20km and is set to 15km otherwise.  For the shape of the vortex outside $R_{max}$, we hold the faster decay length scale fixed at 25km, following the recommendation of Willoughby et al. (2006), and allow the second length scale ($X1$) and the contribution of the fast decay rate ($A$) to vary the shape of the profile. Willoughby et al. (2006) showed a wide range of the slower decay length scale occurs in nature, from

100 km to over 450 km. Given that aircraft data are not uniformly available globally and our intention to only use readily available data (and the difficulties in using aircraft reconnaissance data as discussed in Kepert 2006a,b and Schwendike and Kepert 2008), we decide not to include these additional data sources for subsets of global historical events. Willoughby et

al. (2006) demonstrated dependence on $V_{max}$, and latitude, with the more intense low latitude TCs being more sharply peaked. We therefore choose to allow the remaining two free parameters ($A$ and $X1$) to vary with readily available parameters $R_{max}$, $V_{max}$ and latitude, following equation 11 in Willoughby et al. (2006). This globally consistent approach is needed to allow relative risk assessments across regions.

The Willoughby profile requires $V_{max}$ at gradient wind level. The input track $V_{max}$ is almost universally an estimate of the surface value, so an inflation factor is used to inflate the wind estimate from a surface to a gradient level value. Franklin et al. (2003) observed an inner core wind maximum at about 500m that increases to about 1km for the outer winds. They found a logarithmic profile below the wind maximum and a reduction of winds above due to the warm core. The result is a 700-hPa-to-surface wind factor of about 0.9 in the inner core. Kepert and Wang (2001) theory also has a factor of 0.9 in the inner core, with the

factor decreasing to 0.75 in the outer winds. Knaff et al. (2011) took the Franklin et al. (2003) factor of 0.9 in the inner core and additionally reduced it by a factor of 0.8 to go from marine-exposure winds to terrestrial-exposure winds, giving a net factor of 0.9*0.8=0.72. Initial testing using a variable factor for offshore and onshore track caused enhanced winds just offshore in response to the higher factor for the first inland track point. This is because the outer winds were still responding to the low surface roughness while being driven by stronger gradient winds. We therefore choose to hold the factor fixed at 0.76 (also

based on sensitivity testing, not shown) and appropriate for inland winds.

### 2.3 Stage 2: Modelling the Atmospheric Boundary Layer

A key advance of our modelling approach over traditional approaches is the use of a numerical boundary layer model to generate a surface wind field. Winds in the boundary layer, the layer between the gradient wind level and the surface, are

modelled using a modified version of KW01. KW01 is initialized with the gradient-level wind and pressure fields from the parametric model throughout the entire depth of the boundary layer. It then uses the dry hydrostatic primitive equations (solving for atmospheric flow under conservation of mass, conservation of momentum and accounting for heat sources and sinks) to spin up a steady state boundary layer wind structure in balance with the gradient winds and pressures. Moisture is excluded from the model because of its negligible effects on boundary layer flow. We selected this non-linear model because of its

ability to develop important boundary layer structures such as the super-gradient jet (KW01; Kepert, 2006).

The model rapidly achieves steady state in the strongly forced TC environment characterized by large momentum fluxes and fast adjustments (not shown). The model has 18 vertical levels on a height-based vertical coordinate with the model top fixed at 2.0 km. This height was chosen to be above the height of super-gradient jets, and to be above the typical range of the radially-

dependent boundary layer top (Kepert et al. 2012). This number of vertical levels is far higher than used in most numerical weather prediction models. While the boundary layer height likely varies substantially across global TCs, we choose to keep

this fixed in the absence of readily available data. Sensitivity tests (not shown) show that horizontal grid spacings of 2 to 4 km are sufficient to maintain the tight pressure gradients of strong TCs, and to capture the effects of major terrain features such as coastal ranges or coastal urban areas.

The highly turbulent boundary-layer flow is treated using a high order turbulence scheme with prognostic turbulent kinetic energy and turbulence dissipation, following Galperin et al. (1988). The turbulence length scale is diagnostic and is capped at 80 m following Blackadar (1962). While the model parameterizes shear-driven turbulence, it does not well represent strong thermal effects such as buoyancy. But these thermal effects are negligible for most TC boundary layers where the Richardson number (the ratio of buoyancy-driven to shear-driven turbulence) is close to zero. Harper et al. (2010) state that the outputs of

numerical models without explicit turbulence should be considered to be the mean wind. Our numerically modelled winds are calculated on a timestep of a few seconds. The model can therefore only resolve wind variations of about 4 to 7 times the model timestep, depending on variability of the flow. The instantaneous model outputs are therefore not the instantaneous wind, but closer to the 1-minute mean wind.

The original model coordinates of KW01 are storm-relative. Here, model coordinates are changed to Earth-relative. Each simulation is conducted on one of the 17 geographically-fixed regional domains, shown later in Fig. 5. For the simulation of an entire storm footprint, the forcing of the model from the upper winds and pressure field is updated every 10 minutes. While KW01 found 24 hours was needed for the boundary layer to spin up an equilibrium state, running for 24 hours for each forcing update is computationally impractical. Sensitivity tests (not shown) showed that the surface winds, the most important for this

study, respond rapidly to changes in the forcing.

A code modification allows the boundary layer solution to respond to real-world terrain height and surface roughness as the TC tracks over land. Terrain elevation data are provided by the Global Multi-Resolution Terrain Elevation Dataset 2010 at 30 arc-seconds (Danielson and Gesch, 2011), and are interpolated onto the model grid. Terrain height enters the boundary layer

model through the computation of vertical diffusion and vertical advection, where higher terrain enhances both. Terrain height is first normalized by the height of the model top and capped at 0.9. Vertical motion is diagnosed through the three-dimensional continuity equation integrating upwards given terrain height and horizontal velocity. Mass may therefore enter or exit the model top according to the requirement to balance net horizontal convergence. Land-use roughness is provided by the MODIS-based 21 category land use data at 30 arc-seconds. The model feels the variable surface roughness through the drag coefficient term.

Over land a neutral drag coefficient depends on the surface roughness (Garratt, 1977). Over the ocean, the Charnock relation modified by Smith (1988) is used to account for the effects of increased roughness as wave heights grow with wind speed (see also Powell et al., 2003).

## 3 Case Study: Hurricane Maria (2017)

A series of simulations of increasing model complexity is presented here to illustrate the importance of variable land surface friction and terrain height. Using the case study of Hurricane Maria (2017) over Puerto Rico we first compare simulations using the Willoughby profile only, and the addition of KW01. All simulations were run at 2 km grid spacing. The Willoughby profile (Fig. 2a) places the strongest winds to the right of track, as expected. It captures the decaying winds as Maria crosses Puerto Rico but it misses any abrupt changes in the onshore and offshore flow, also as expected. In the absence of changing surface friction and terrain at landfall, the only information the model has about landfall is through the decreasing $V_{max}$ in the input track data. The addition of the boundary layer model (KW01, Fig. 2b) brings an overall reduction of the footprint (compare Figs. 2a and 2b) and much greater small-scale variability in the footprint over land. Variable terrain height results in wind acceleration over elevated terrain. Variable surface roughness results in sharp transitions in the wind speed along coastlines and, for example, over the urban area of San Juan in the northeast of mainland Puerto Rico. The winds weaken abruptly as Maria makes landfall and the boundary layer model adjusts to the increased surface friction of the land surface. Maximum values of the footprint in the vicinity of the track agree reasonably well with the input track $V_{max}$ values (shown by the coloured dots along the track). The surface reduction factor (the ratio of Fig. 2a and Fig. 2b), shows strong spatial variability (Fig. 2c). The reduction factor ranges between 0.5 and close to 1.0 according to the spatial surface roughness and spatial terrain height.

A snapshot of the simulation using the Willoughby profile and KW01 at the time of landfall is shown in Fig. 2d. Onshore winds decay over land in response to the enhanced surface roughness, overland winds accelerate over elevated terrain, and offshore winds accelerate over the water. Interestingly, wind vectors suggest the high momentum air is carried directly over local terrain features with no evidence of flow deviations or blocking (upstream deceleration).

A scatter plot of model (using the Willoughby profile and KW01) versus observed 1-minute wind speeds throughout the lifetime of the storm at the locations of surface observing stations is shown in Fig 2e. Observations are provided by the 3-hourly NMC ADP Global Surface Observations Subsets (NCEP/NWS/NOAA/U.S. Department of Commerce). Wind averaging periods are converted from two-minute to one-minute for onshore station data and from 10-minute to one-minute for offshore buoy using the World Meteorological Organization conversion factors in Table 1.1 of Harper et al. (2010). Comparisons between observations and model are made using model time within 5 minutes of the observation time. We choose not to adjust for the complex exposure differences across the observing sites or between the observing sites and the model exposure representation.

Differences between model and observations are mostly within $\pm 10$ ms$^{-1}$ across Puerto Rico (Fig. 2e). While a perfect correspondence between model and observations would lie along the one-to-one line, some scatter is expected due to the

relatively coarse model grid not resolving fine-scale variability and the loss in predictability of fine-scale variability. While there is evidence of a small high wind speed bias, particularly for low wind speeds, we choose not to tune the model to a single storm. An evaluation over a collection of storms is presented in the next Section.

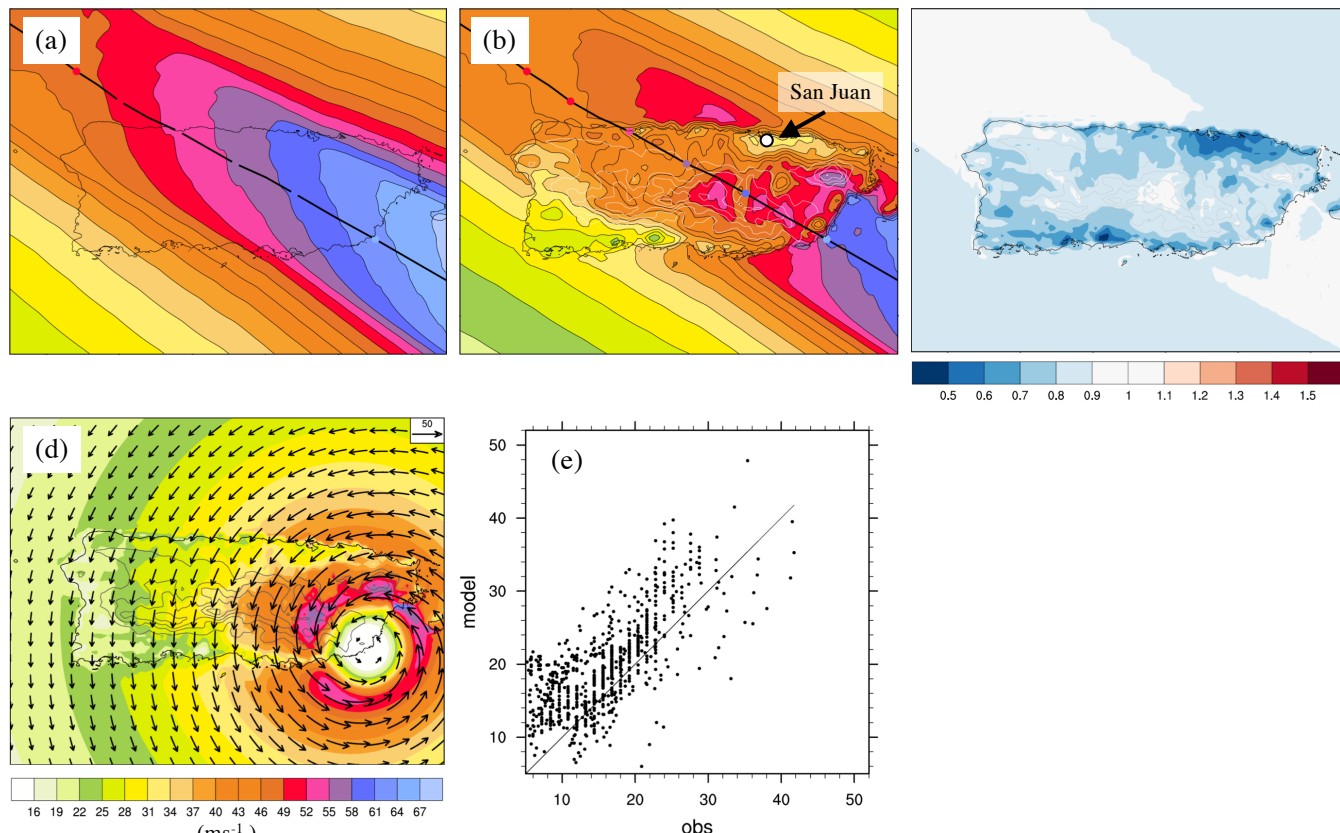

**Figure 2: Case study simulation of Hurricane Maria (2017) over Puerto Rico. Simulated footprints (ms⁻) are shown using (a) Willoughby only, (b) Willoughby and KW01. The hurricane track is shown by the thick black line with input $V_{max}$ shown every six hours along the track (coloured dots). Coastlines are shown by the thin black lines, and are only included in (a) to aid interpretation. (c) The ratio of Willoughby and KW01 to Willoughby only. (d) A snapshot of the simulation using Willoughby and KW01 at the time of landfall (wind speed is contoured and wind vectors are shown in arrows). Terrain height is contoured every 200 m in (b) – (d). (e) A comparison of observed and simulated (Willoughby and KW01) winds.**

The effects of variable surface roughness and terrain height are explored in detail through a series of sensitivity simulations, again using the case of Hurricane Maria (2017) over Puerto Rico. These sensitivity simulations all use the Willoughby profile and KW01 but differ in the representation of the land surface. The representations are i) no land (entire domain set to water, referred to as NO_LAND) to isolate the effect of adding KW01 to Willoughby, ii) no terrain height (entire domain is flat,

referred to as NO_OROG) to isolate the effect of adding variable surface roughness, and iii) all surface roughness set to open water values but retaining terrain height (referred to as NO_ROUGHNESS) to isolate the effect of variable terrain height.

NO_LAND (shown in Fig. 3a) shows a spatially smooth reduction in wind speeds compared to the simulation using the Willoughby profile only (compare with Fig. 2a). The reduction factor is approximately 0.9 (Fig. 3d). NO_OROG (Fig. 3b) shows the strong frictional effect of the drag of the land surface on the boundary layer winds. The wind reduction factor (relative to NO_LAND) falls below 0.6 over the roughest terrain (the variable roughness is shown in Fig. 3g). Finally, NO_ROUGHNESS shows wind acceleration over elevated terrain (see Figs. 3c and 3f). The wind enhancement factor (relative to NO_LAND) increases with terrain elevation but does not exceed 1.3 across Puerto Rico.

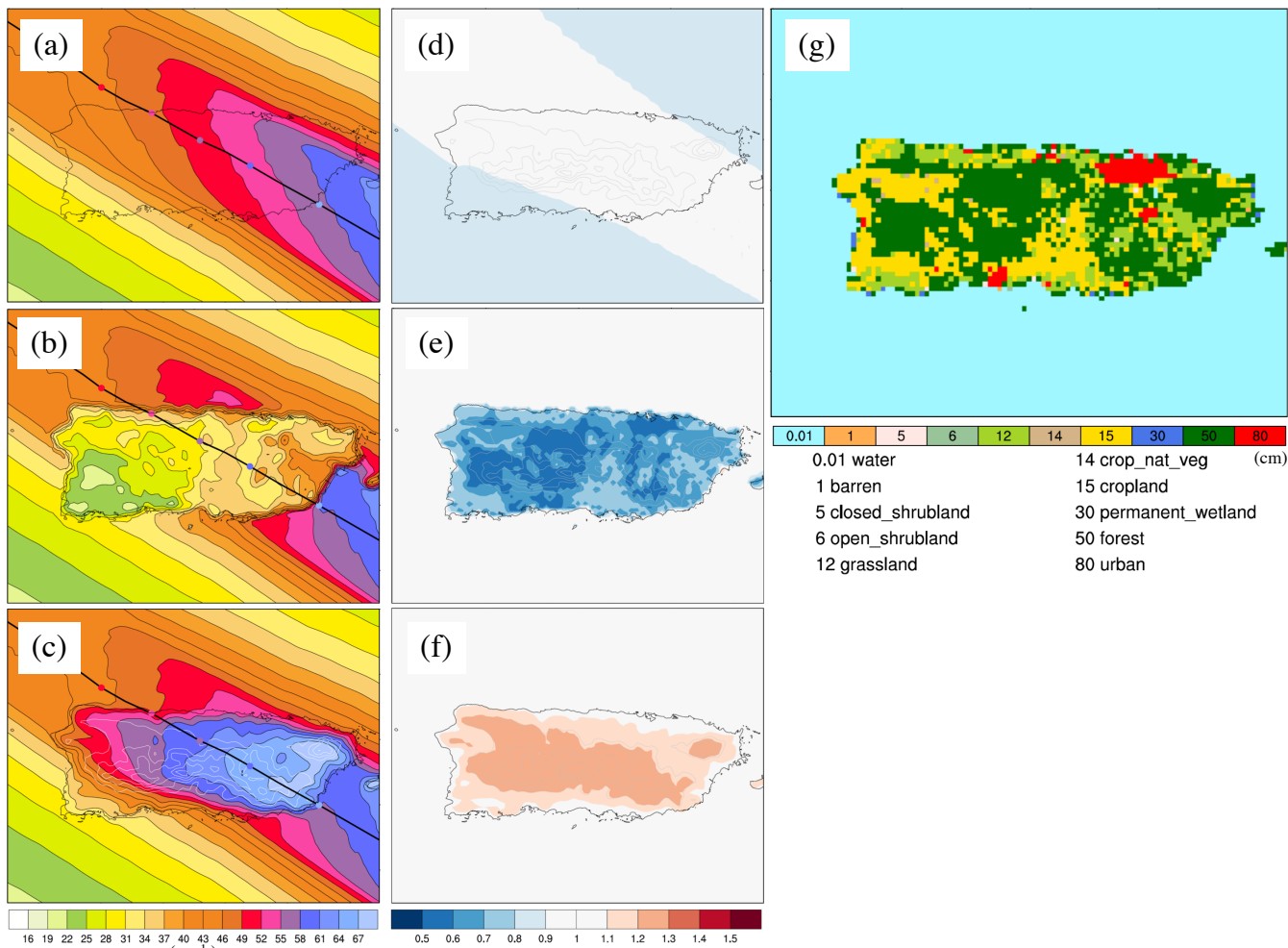

**Figure 3: (a) – (c) Simulated footprints (ms⁻¹) of Hurricane Maria (2017) over Puerto Rico using Willoughby and KW01 with (a) no land (NO_LAND), (b) no terrain height (NO_OROG), and (c) constant surface roughness equal to the ocean value (NO_ROUGHNESS). The hurricane track is shown by the thick black line with input $V_{max}$ shown every six hours along the track**

(coloured dots). Coastlines are shown by the thin black lines, and are only included in (a) to aid interpretation. (d) Ratio of Willoughby only to NO_LAND. (e) Ratio of NO_LAND to NO_OROG. (f) Ratio of NO_LAND to NO_ROUGHNESS. (g) Surface roughness (cm) of the land-use categories.

## 4 Evaluation

Kepert (2012) notes that "The boundary layer in a tropical cyclone is in some respects unlike that elsewhere in the atmosphere. It is therefore necessary to evaluate boundary layer parameterizations for their suitability for use in tropical cyclone simulation." Here we present a model evaluation for a subset of historical landfalling TCs to assess the model's capability to reproduce observed surface wind speeds.

While reanalysis products provide historical footprints as a convenient gridded product, they themselves are a modelled product that contains various assumptions and inaccuracies. In addition, reanalyses are typically standardised to a given land surface type. The HWIND reanalysis product (Powell et al., 1998), for example, is only valid for open-terrain exposure and therefore commonly exceeds wind values from surface observing stations. We therefore choose to evaluate the model against the surface station observations provided by the 3-hourly NMC ADP Global Surface Observations Subsets
(NCEP/NWS/NOAA/U.S. Department of Commerce). Since the U.S. has the highest density observing sites, a subset of eight U.S. landfalling storms was chosen for the evaluation. This subset includes storms making landfall on the Gulf Coast (Rita (2005), Katrina (2005) and Ivan (2004)), Florida (Charley (2004), Irma (2017) and Wilma (2005)), and the U.S. Northeast (Sandy (2012) and Irene (2011)).

Model performance across the eight U.S. storms is summarized in Fig. 4. To better understand model performance, the comparison with observations explores model bias as a function of distance from the TC centre, and split by left-of-track and right-of-track. Comparisons between observations and model are made using model time within 5 minutes of the observation time. Figures 4a and 4b show there is little evidence of large bias with the vast majority of differences falling within ±10 ms$^{-1}$. There is also no strong variation in bias with distance from the storm centre. A possible explanation for an apparent low bias
within 20 km of the storm centre is storm centre location error in the input track data. Our approach compares favourably with a recently-published global modelling approach. The approach of Tan and Fang (2018) that combines parametric wind profile modelling with local wind multiplication factors produces typical errors of 8 to 10ms$^{-1}$ in the 10-minute mean wind. In addition, this magnitude error can also be present in hurricane surface wind vectors utilizing C-band dual-polarization synthetic aperture radar observations, when compared to collocated QuikSCAT-measured wind speeds (Zhang et al. 2014).


The model performs similarly well on both sides of the storm, indicating that our treatment of asymmetry due to translation speed captures a major portion of the observed asymmetry. Figures 4c and 4d show only urban locations that observed winds exceeding 18 ms[1]. While there is a suggestion of a low bias for winds far from the storm centre, the vast majority of points lie within 10 ms[1] of the observations. The most damaging winds also reside close to $R_{max}$ (in the range 20-100 km from the TC

5   centre) where our bias is smallest. Holmes (2007) found that the roughness length for urban areas can vary between 0.1 and 0.5 m for suburban regions and rise to between 1 and 5 m for densely packed high-rises in urban centres. Our model uses a single roughness length for all urban areas (suburban and city centres) of 0.8m, taken from the MODIS land use dataset. This value may be too high for suburban areas, where a value closer to 0.2 is typical (Yang et al. 2014). Depending on the specific siting of the wind observing stations, it's probable that the introduction of multiple urban categories with different roughness

10   lengths would improve our low wind speed bias.

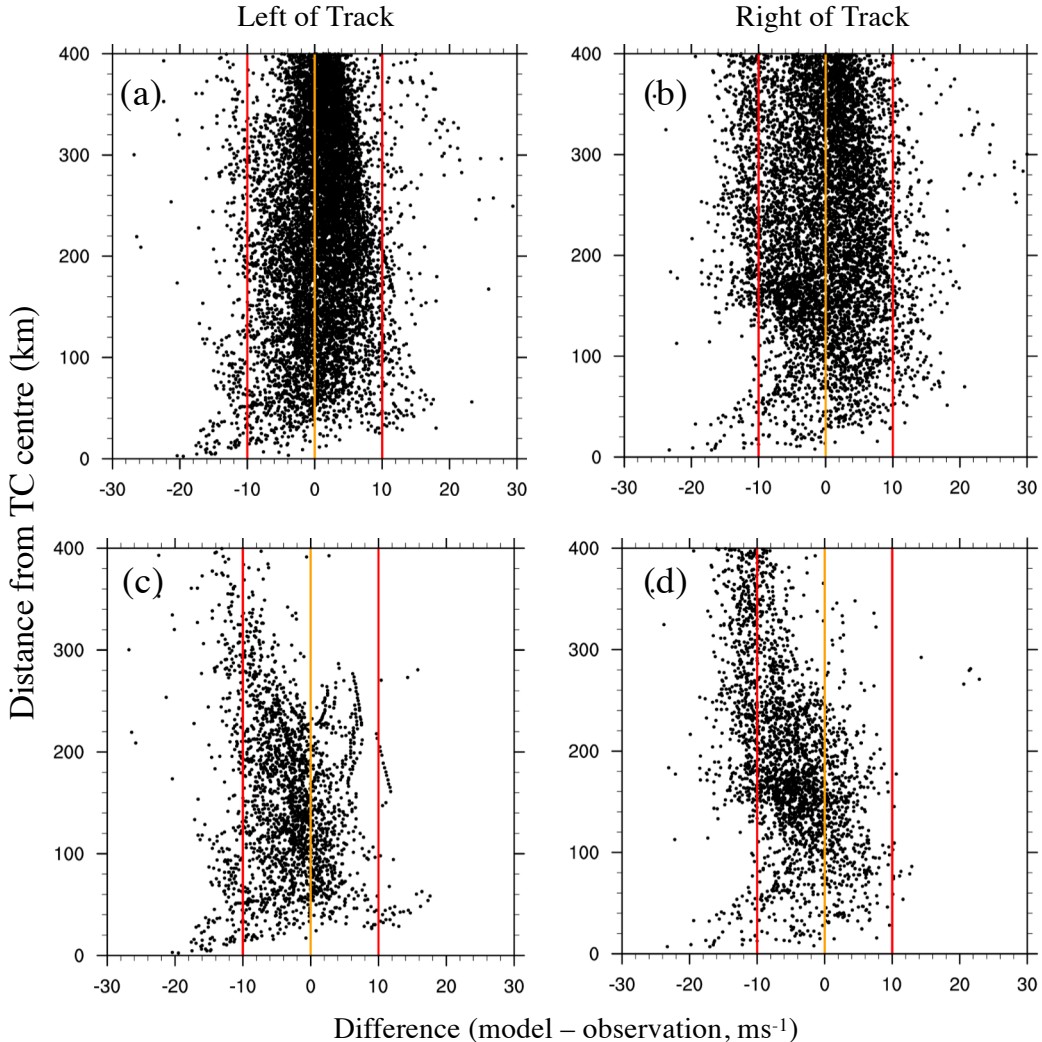

**Figure 4: Difference between modelled and observed wind speeds for the 8 U.S. storms as a function of distance from the TC centre (ms⁻¹) and split by left of track (left column) and right of track (right column). The upper row shows all data points, and the lower row shows only urban data points that experienced observed wind speeds greater than 18 ms⁻¹. The orange lines indicate zero difference and the red lines indicate a positive and negative difference of 10 ms⁻¹.**

### 5. A Dataset of Global Historical Landfalling TC Footprints

One application of the model is demonstrated here through the creation of the dataset of global historical landfalling TC footprints. The dataset consists of 714 footprints. Figure 5 shows the locations of 17 simulation domains together with the numbers of simulated footprints per domain. Figure 6 shows all tracks simulated for three example domains: The Gulf and Southeast U.S. coast, Eastern China and Taiwan, and Eastern Australia. The numbers and spatial density of tracks vary due to

different periods of records for the different basins and the different frequencies of landfalling storms. Each footprint contains the storm lifetime maximum 1-minute average wind at each grid point at 10 meters above Earth's surface. The units are meters per second. Each footprint is on a latitude longitude grid with a grid spacing between 2 and 4 km depending on the regional domain.

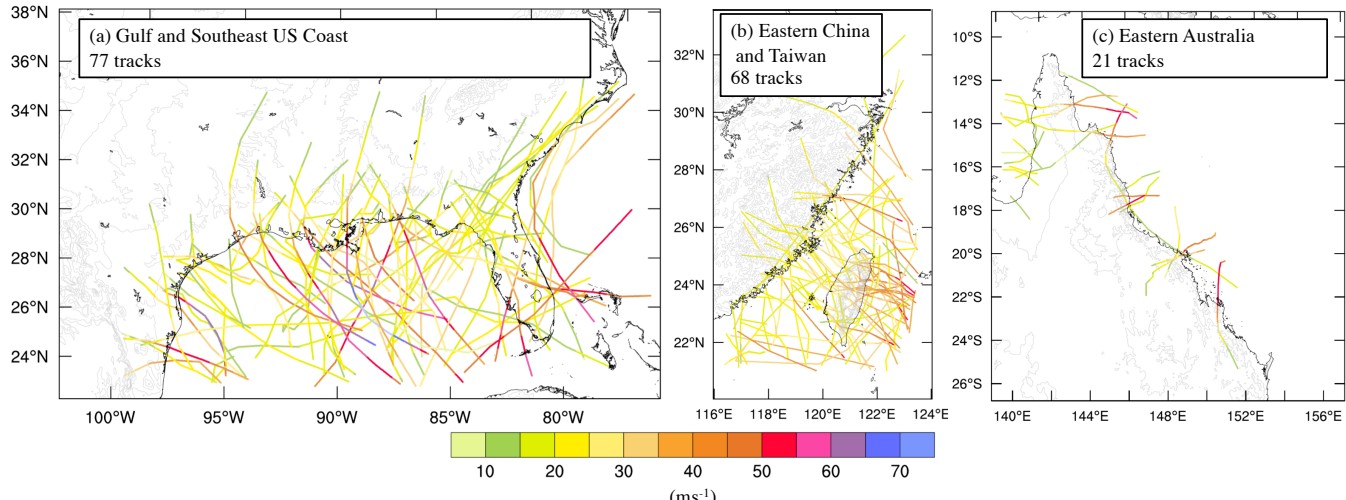

**Figure 5: A global map of the 17 simulation domains used in the creation of the dataset of historical global TC footprints. The data record length extends as far back as the required input data are available. Archived $R_{max}$ data extend back to 1988 for the North Atlantic and the East Pacific, but extend back only as far as the early 2000s for the other basins. The numbers of simulated footprints for each domain is indicated.**

**Figure 6: TC tracks used to simulate footprints for domains over a) the Gulf and Southeast U.S. coast, b) Eastern China and Taiwan, and c) Eastern Australia. Tracks are coloured by the track Vmax (ms-1). Track data are taken from IBTrACS (Knapp et al. 2010).**

Tan and Fang (2018) suggest substantial regional variations exist in the inland extent of strong wind. A preliminary analysis of regional variability in the wind speed decay rates with inland track distance is presented here. Given that the simulated gradient winds are driven by the input best track data, the cyclone-scale inland decay is included to the extent it is included in the best track data. Additional sub-cyclone-scale terrain effects are included through the interaction of KW01 and the surface.

Figure 7 shows the regional average along-track distance rate-of-change of storm lifetime maximum wind speed and terrain height with along-track distance from the point of landfall for the Gulf and Southeast U.S. coast, Eastern China and Taiwan, and Eastern Australia (the same regions as shown in Fig. 6). The wind data are extracted from each wind swath at the location of the TC track. This data is therefore the storm lifetime maximum wind speed at specific locations along each TC track. All along-track wind data for a given region are then composited about their points of landfall, giving the region-average along-

track wind swath vs. distance inland. All data are additionally smoothed using a 30 km running average. The strength of the smoother was chosen as a balance between the need to smooth noisy wind profiles while retaining the effects of coastlines and terrain. The x-axes (along-track extent) extend until the distance inland at which only three tracks remain.

For the Gulf and Southeast U.S. region averages are calculated over 77 tracks. We see two regimes of behaviour (Fig. 7a). The

winds strongly decay at the coast as the boundary layer adjusts to the increased surface roughness. The winds then decay more moderately as the tracks extend further inland. The average along-track terrain gradient gradually rises to a peak of 210 m at an along-track distance of 550 km from the point of landfall and does not appear to substantially affect the inland wind profile. For other regions, however, steeper orography appears to have a large effect on the inland winds.

Figure 7b shows the average along-track winds calculated over 68 tracks over Eastern China and Taiwan. The average landfall wind speed of 29 ms⁻¹ experiences an abrupt decay at the immediate coast, followed by a modest recovery as the storms pass over the steep windward slopes of Taiwan and mainland China. The winds then experience some of the strongest decay rates in the entire dataset on the lee-side. The distance rate-of-change of wind speed is the net effect of orography, surface roughness and the overall inland decay according to the input best track data. This makes it challenging to isolate the processes driving

the inland wind speed gradient. It's possible that the increasing along-track terrain height drives enhanced vertical diffusion and vertical advection in the boundary layer model, and enhanced horizontal flow through the three-dimensional continuity equation. But idealized modelling would be needed to identify the presence and strength of this proposed mechanism.

The strong influence of terrain is also seen along the coastal ranges of Eastern Australia. Figure 7c shows three peaks of

between 250 m and 280 m in the average terrain height along 21 tracks within 300km of the average point of landfall. Again, the average landfall wind speed of 28 ms⁻¹ experiences a strong wind decay at the immediate coast before recovering slightly over the first ridgeline and then strongly decaying on the lee side. The rate of decay lessens over the second and third ridgelines. Overall, the relationship between wind change and terrain height does not appear to exhibit lead-lag behaviour.

This preliminary analysis suggests a strong influence of regional terrain on overland footprints. Further investigation is needed to better quantify the effect and understand the extent to which the full range of terrain effects on TC wind fields are captured by this modelling approach.

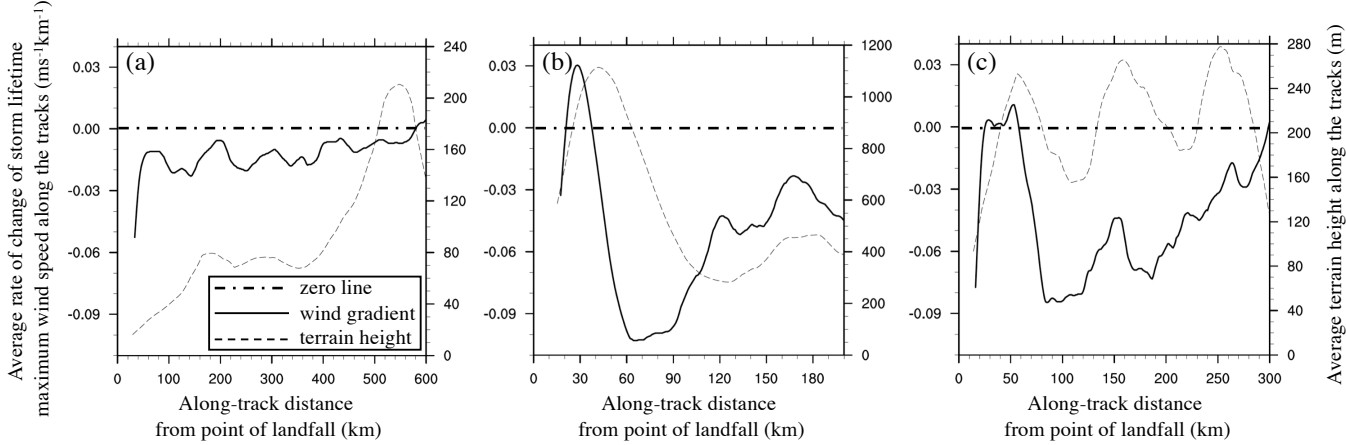

Figure 7: Variation of regional average distance rate-of-change of wind speed (ms⁻¹ km⁻¹) and terrain height (m) with along-track distance from the point of landfall (km) for the same regions as shown in Fig. 6; a) Gulf and Southeast U.S. coast, b) Eastern China and Taiwan, and c) Eastern Australia. All data are smoothed using a 30 point running average. Region average values are calculated over all tracks within each region. The x-axes (along-track distance) are cut off at the point where only three tracks remain. Each panel has the same left y-axis limits but different right y-axis limits to better show the ranges of regional terrain height.

## 6. Conclusions

This paper presented a novel and globally-applicable approach to modelling the surface wind field of landfalling TCs. The modelling system simulates the temporal evolution of the near-surface spatial wind fields of landfalling TCs, accounting for terrain effects such as coastlines, inland orography, and abrupt changes in surface friction. A two-step process models the upper wind field using a parametric wind field model fitted to TC track data, then brings the winds down to the surface using a numerical boundary layer model. This represents more of the boundary layer physics and physical terrain effects than analytical approaches or empirical wind reduction factors. The guiding principles for model development were to i) use only readily available track data from historical archives, real-time forecasts or synthetic track models, and ii) maintain balance between representing the necessary physics of the land surface-flow interactions and the need for computational speed for future applications to probabilistic wind speed assessment.

The model is suitable for simulating the near-surface wind field throughout the entire lifecycle of translating, strengthening/weakening, expanding/contracting, and landfalling TCs.

An evaluation of a subset of eight U.S. landfalling TCs against surface station observations showed that the model had no large bias across all storm radii, and across both sides of the storm tracks.

For a case study of Hurricane Maria (2017), the inclusion of variable surface friction and terrain height was shown to add substantial sub-storm scale variability to the footprint. Winds dropped abruptly at the coast, yet accelerated over windward slopes and mountain crests. Winds also decelerated over the high surface drag of urban areas. The gradient-level to 10-meter reduction factor ranged between 0.5 and close to 1.0 according to the spatial surface roughness and spatial terrain height. Separating the surface roughness and terrain height effects showed surface roughness factors can fall below 0.6 whereas terrain height enhancement factors did not exceed 1.3. Analysis of wind vectors suggest that high momentum air is carried directly over local terrain features with no evidence of flow deviations or local blocking. Our modelling approach does not allow any blocking to affect the TC track itself, although the observed track data do include such blocking effects.

Further work is needed to verify the extent to which the full terrain and surface drag effects are included in the modelling approach. In addition to a process-level evaluation against observations, the efficacy of the approach could be assessed through comparison with numerical weather prediction (NWP) model simulations to understand where the approach fails. But differences in observed and NWP simulated TC tracks would need careful consideration. The overarching aim would be to identify the key terrain effects needed to be included in computationally efficient overland TC wind models.

An evaluation of a subset of eight U.S. landfalling TCs against surface station observations showed that the model had no large bias across all storm radii, and across both sides of the storm tracks. Our approach compares favourably with a recently-published global modelling approach and differences between different observing systems. Considering only urban locations that had observed winds greater than 18 ms[1] there is a suggestion of a low bias. This may be due to our use of a single urban roughness length of 0.8m that is high compared to other studies.

The challenge of developing a globally applicable approach is that the accuracy at the individual event level will be lower than for a model developed for individual events or specific regions. We therefore do not expect our approach to improve upon individual event-level approaches that assimilate additional observational data. But a globally applicable approach has a number of unique benefits that derive from its small amount of required input data and physical response of the boundary layer winds to terrain. These benefits include generating events in data sparse-regions, generating synthetic events, and application to downscaling TC tracks from global climate models

An application of the model was demonstrated through the creation of a dataset of 714 global historical TC footprints, and is referred to as the Willis Research Network Global Tropical Cyclone Wind Footprint dataset version 1. While previous studies have mapped global historical TC wind fields, none included the nonlinear adjustments of the surface wind field to variable

terrain. This unique dataset is a rich resource to advance our process-level understanding of spatial and temporal variability in overland TC winds. A preliminary analysis showed strong regional variability in the inland extent of damaging surface winds, as controlled by regional TC and terrain characteristics. Analysis of regional average footprints showed acceleration over windward slopes leading to some recovery of the abrupt wind speed reduction at the immediate coast. For risk

management, this dataset may be used to better understand historical losses in regions of complex topography, and support the generation of synthetic event sets, particularly in regions of sparse historical data.

For large domains needed to capture the long tracks of fast-moving storms - over Japan or the Northeast U.S., for example - the simulation wall-clock time is substantially longer than using wind profile models alone. The most expensive domain, over

Japan, at 900 x 1100 x 18 grid points using a 2-second timestep, a 24-hour simulation takes 6 hours wall-clock time on 36 cores. Smaller domains at a coarser 4-km grid spacing run much faster. This is efficient compared to the costs of high-resolution NWP simulations and therefore offer a computationally feasible approach to explore wind risk in complex terrain, while acknowledging that NWP models capture a fuller representation of terrain effects.

Other applications include real-time forecasting of overland TC winds in advance of approaching TCs. The model also may be used to produce wind exceedance probabilities (following a similar approach to that presented in Arthur (2019) and Arthur et al., 2008). High efficiency, relative to numerical weather prediction (NWP) simulations, permits large numbers of simulations that could be used as inputs to a Generalized Extreme Value fit to the data to quantify the extremes. Another opportunity presented by this three-dimensional modelling of the boundary layer wind structure is an assessment of wind loading on high-

rise structures. Today's coastal high-rise structures can extend above the surface layer into wind speeds far in excess of those at the surface (Vickery et al., 2009) at heights that are explicitly simulated in the model.

While the modelling approach captures more of the dynamics and physics of the TC boundary layer than analytical or empirical approaches, it misses a number of potentially important processes. A nonhydrostatic modelling system, for example, would

capture more of the orographic effect (Wang, 2007). Perhaps more important is its accounting for only one-way of what is inherently a two-way interaction between the boundary layer and the free troposphere. For example, terrain variations can enhance convergence and trigger deep convection that may feedback on the low-level winds. TC responses to changes in land surface, such as at landfall, can have substantial effects on the whole TC circulations (e.g., Ramsay and Leslie, 2008; Wu, 2001). Another limitation is the use of a parametric TC wind profile model that is not designed to fit wind profiles of extra-

tropical transitioning cyclones. During the process of transition, the wind field can become highly asymmetric and develop wind maxima on either side of the cyclone and far from the cyclone centre (e.g., Loridan et al., 2015). This presents a limitation of our modelling approach and may cause substantial errors in the footprints of strongly transitioning TCs over higher latitudes of the U.S. and Japan, for example. Finally, for wind loading and risk management application, an explicit representation of gusts is desirable.

This paper demonstrates the potential benefits of using a parametric wind-field model with a physical representation of terrain effects to model overland TC surface winds. Future work will explore the added value of this approach compared to the use of local wind multiplication factors. A more detailed process-level evaluation of terrain effects will identify the extent to which physical terrain effects are represented by this approach. Additional experiments should also explore sensitivity to model parameters such as model top height. Finally, future work should assess the value of this modelling technology and global landfalling TC catalogue in risk management decision making contexts.

## Author Contribution

JD, GH, IDW, SP and GS designed the investigation. YW led the methodology and software with contributions from JD and MG. MG ran the simulations, formal analysis, visualization and data curation. JD prepared the writing with contributions from all co-authors.

## Competing Interests

The authors declare that they have no conflict of interest.

## Data Availability

The Willis Research Network Global Tropical Cyclone Wind Footprint dataset version 1 will be made publicly available on the lead author's GitHub site on 1ˢᵗ May 2020. This time restriction follows terms of the funder, the Willis Research Network.

## Acknowledgements

This work was funded by the Willis Research Network. The visit of Dr. Yuqing Wang to NCAR for the boundary layer model setup was partly supported by NCAR Mesoscale and Microscale Meteorology laboratory visitor funds. NCAR is sponsored by the National Science Foundation. We thank two anonymous reviewers and Bruce Harper for comments that greatly improved the manuscript.

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
