# Peer review of "Modelling Global Tropical Cyclone Wind Footprints"

_Natural Hazards and Earth System Sciences, 2019_

## Referee Comment (RC1) · Anonymous Referee #1 · 8 Aug 2019

The manuscript proposed a new modeling system for generating tropical cyclone (TC) wind, which consists of a parametric radial profile model, the non-linear boundary layer model (KW01), and the terrain effects. The authors presented a case of hurricane Maria and Wilma and verified the model using landfalling storms. Then, the authors discussed the impact of terrain on the changes of TC winds over the South East US, Taiwan and Eastern China, and Eastern Australia. Overall, I like this approach and can think of many applications of this model. One criticism I have is that the advantage of using KW01 was not shown or discussed with evidence. In Section 2.3, The authors described KW01 as the key advancing component of the modeling system. At many other places, the author says that their system contains more dynamical processes than other existing tools. However, none of the results shown here can isolate the positive impact of using KW01. I suggest conducting additional simulations using Willoughby's wind with an empirical factor applied to get winds at10m height, plus the

terrain effect. By comparing these new simulations with the Willoughby+KW01+terrain, we can see the advantage of (or differences caused by) KW01.

Below are a few minor comments and questions

1. Page 4, line 2. While I understand the advantage of using KW01 instead of a simple empirical model, this sentence sounds vague. Please elaborate more on what the additional dynamical effects are.

2. The first paragraph of section 5 (page 11) should belong to Section 2.1.

3. Page 5, Line 15: Please mention the TC boundary layer height used in this study as well. Is the model performance sensitive to the TCBL height?

4. Page 7, L11 'running for 24 hours for each forcing update is computationally impractical.' I am surprised to see that running 24 hours of KW01 is computationally impractical. What is the computational cost of KW01, and how is it compare to the computational cost of 2-km WRF.

I am asking this is because, for assessing wind risk, the most significant advantage of a simplified wind generator v.s. a full-physical model is its low computational cost. If running KW01 is computationally expensive, this system will not be able to use for real risk assessment, which (I thought) is one (and probably the mostÂăimportant one) of motivations of this work. (The other motivation is to understand wind risk over complex terrain using historical cases. For this purpose, we can always run WRF or other mesoscale models which may generate more realistic winds than KW01)

5. Page 9, L24. Do you mean the maximum wind speed recorded at the station during the lifetime of the storm, which is different from the storm lifetime maximum wind speed (which is usually one value per storm)?

6. Page 11, L1: Where is this 20% bias correction factor comping from? Is it universally applied to all simulations?

7. Page 11. L12-14 belongs to the figure caption of Fig. 6, not in the main text.

8. Figure 7 and the related discussion. Did you check the enhanced vertical diffusion and vertical advection in KW01? Can you show some analysis of these enhanced features? There is a lag between the terrain and the wind gradient. Why?

9. Willoughby et al. 2006 is missing in the references.

---

## Referee Comment (RC2) · Anonymous Referee #2 · 12 Aug 2019

See attached file

Please also note the supplement to this comment:
https://www.nat-hazards-earth-syst-sci-discuss.net/nhess-2019-207/nhess-2019-207-RC2-supplement.pdf

---

## Short Comment (SC1) · 12 Aug 2019

Summary

The authors seek to "to advance our understanding of overland wind risk in regions of complex terrain and support wind risk assessments in regions of sparse historical data" through the application of a combination of analytical, numerical and empirical techniques. The authors make reference to many very successful more simplified approaches that have been developed over the past three decades that emanate from wind engineering, atmospheric science and insurance loss initiatives. They put the case that previous approaches, in their assessment, lack the essential capacity to incorporate complex terrain and "the essential dynamics and physics" of tropical cyclone (TC) behaviour where "the accuracy of wind speeds over urban (sic) is of critical importance". The use of a diagnostic 3D numerical boundary layer model is central to their

thesis.

While the desirability of such an approach can be supported, where practical, it is ironic that the demonstrated model skill is so poor at reproducing recent historical TC winds in areas where there is a significant amount of quality data available and in mostly flat landscapes. In no way does the model "compare favourably" with the displayed data. An 8 to 10 m/s error band in any hindcast event intended to assist, for example, offshore engineering design or sensitive onshore high-rise design, would be regarded as completely unacceptable. For insurance losses where damage is noted by the authors to be additionally highly nonlinear with wind speed, it would be massively unreliable. The stated need to apply an empirical "20% adjustment for urban areas" is not only completely inconsistent with the theoretical high ground being argued but is symptomatic of a modelling system that has some significant problems.

The Challenge of a Global Approach and the Expected Benefits

The problem with developing tools for global application unfortunately means that accuracy is inevitably impacted by the need to adopt spatially and/or temporally compromised globally available datasets. This situation limits, and actively dissuades, examination of the myriad of fine scale site-specific influences on the TC surface wind during a specific event that cannot be ignored. These have traditionally been transparently handled by reference to standard exposure and application of statistically based boundary layer turbulence approaches. The authors' more complex and computationally demanding approach needs to demonstrate at least a comparable utility.

In any case, it is not clear what practical application there is in producing such a global (deterministic) event set, given that the essential need is for risk management that implicitly requires a probabilistic approach. Cherry-picking of historical events does not yield firm statistical guidance and any such results will most likely be less reliable than any regional wind speed risk assessment that is based on (even sparse) long term data sets, given that aggregation of sites is often justified. In spite of the practical challenges

in this space, the various engineering design standards around the world have over-time assembled realistic and likely suitably conservative wind risk frameworks, all of which embody the need to allow for height, terrain and topography effects.

The model's development is touted as valuable for insurance-related purposes. However, the principal cause of increasing world-wide losses for insurers is, together with uninformed risk-based planning, the failure to implement known good design, construction and inspection practices for residential development. The importance of globally modelling large scale terrain influences is also overstated given that, compared with typically nearly-flat or undulating conurbations, there is negligible insurance exposure to wind hazard in areas of very high or steep terrain.

Comments on Method

(2.) The step that "removes an estimate of the asymmetry due to storm motion" from the surface wind relies on the assumption that historical Vmax do reliably include such an influence. While Dvorak, for example, implies that is the case there is no specific allowance in the methodology. Hence the adopted empirical adjustment likely has little merit in terms of overall accuracy.

In quoting Harper, Kepert and Ginger (2010) (aka the WMO wind averaging guidelines - hereafter HKG), the assumption that numerically modelled winds calculated at a small timestep are representative of so-called 1-min sustained winds is incorrect. Section 1.6 of HKG specifically advises on that topic noting that numerical models without explicit eddy representation only estimate mean wind speeds, not gusts such as the so-called 1-min sustained wind. However, correcting for that (e.g. per HKG Table 1.2) will likely have no specific effect on the model performance.

(2.1) The authors state that the model is "agnostic to the source of the track data" as though that is some advantage, whereas the vast majority of historical datasets consist only of (lat, lon, Vmax) estimates with acknowledged high variability between agencies and also over-time. Rmax is also noted to be an essential parameter but

is only recently available in some regions and not transparently derived. While these drawbacks are unavoidable, it further emphasises the challenge of using any (global) historical track data without critical assessment and expecting a high level of accuracy in the estimation of terrain-sensitive surface winds.

(2.2) While the Willoughby profile may be superior to some others for a hands-off global application it still requires an outer scale assumption and to note that TC scale, and its temporal evolution, is a critically important parameter in accurately modelling surface winds. The adopted land use surface roughness, with a scale of about 1 km, seems reasonable enough but should also be verified by example for the set of modelled storms. To note also that the references cited for drag coefficients are very dated and the authors could adopt more recent evidence to better suit their argued approach.

Comments on Results and Evaluation

(3.) The Puerto Rico example illustrates the intention and ability of the model to introduce terrain and topographic variability into its results but, without any verification, is otherwise meaningless and simply an "artist's impression".

(4.) To note that the use of 3-h sampled wind data is typically inappropriate for TC passages within, say, 100 km and may be a principal source of the poor comparisons. Application of HKG Table 1.1 is also dubious given the 3-h sampling but more so because of its limited and nominal exposure classes, which appear inconsistent with the aim of deriving fine scale surface winds. HKG Section 1.2 says "The aim has been to provide a broad-brush guidance that will be most useful to the forecast environment rather than a detailed analytical methodology" and "In particular, post analysis of TC events should seek to use the highest possible site-specific analytical accuracy for estimating local wind speeds. This would include consideration of local surface roughness, exposure and topographic effects when undertaking quantitative assessments of storm impacts." This implies an approach like Powell et al. (1996) is needed in such cases. Again, these oversights in applying HKG will likely have little effect on model

performance, but do point to a lack of rigour in matters of wind magnitude adjustment.

As noted previously, the demonstrated performance of the model is very poor compared with the numerous less-complex examples that are cited. In spite of the authors' tendency to downgrade the utility of H*WIND I would instead encourage pursuing such comparisons in order to locate the model deficiencies and improve its performance for the demonstration storms.

Comments on Global Landfalling TC Footprints

(5) The detailed commentary and interpretation of aspects of modelled landfall and inland wind decay characteristics in various localities seems to overlook the fact that the model is only reacting to the imposed "best track" intensity variation and therefore can have no better skill than offered by a simpler parametric approach. Surely fully dynamic modelling (e.g. HWRF or similar) is needed to reliably explore such impacts.

---

## Author Comment (AC1) · 4 Nov 2019

**Dear Bruce Harper**

Thank you for your in-depth and informed comments. Our responses to your comments and the comments of the other two reviewers will greatly improve the manuscript. Below are our responses (in blue) to each comment in turn.

While responding to comments from all reviewers we found a bug in our code in the way we treat asymmetry. As described in the original manuscript, we first remove an estimate of the asymmetry due to forward speed from the input best track  $V_{max}$ . The portion removed is a function of the TC translation speed,  $V_a=1.173V_1^{0.03}$ , following Chavas et al., (2017). We then add back an estimate of the asymmetry to the spatial 10m wind field diagnosed by KW01, again following Chavas et al., (2017). And in addition, we apply a factor that varies with radial distance from the storm center (the factor is equal to 1 at the radius of maximum winds and then decays with increasing radius) following Jakobsen and Madsen (2004). The bug was that the code missed adding back the  $V_c$ -dependent factor and only added back the radially-dependent factor. This caused us to add back the full value of  $V_c$  at  $R_{max}$  which caused too strong asymmetry, particularly for fast moving storms over Japan for example. In response, original figures 2, 3 and 4 will be corrected. The analysis in original figure 7 is for along-track winds and so will not be affected by this bug fix.

Chavas, D. R., Reed, K. A. and Knaff, J. A.: Physical understanding of the tropical cyclone windpressure relationship. Nature communications, 8(1), p.1360, https://doi.org/10.1038/s41467-017-01546-9, 2017.

Jakobsen, F. and H. Madsen, H.: Comparison and further development of parametric tropical cyclone models for storm surge modeling. Journal of Wind Engineering, 92, 375-391, https://doi.org/10.1016/j.jweia.2004.01.003, 2004.

**Bruce Harper**

The authors seek to "to advance our understanding of overland wind risk in regions of complex terrain and support wind risk assessments in regions of sparse historical data" through the application of a combination of analytical, numerical and empirical techniques. The authors make reference to many very successful more simplified approaches that have been developed over the past three decades that emanate from wind engineering, atmospheric science and insurance loss initiatives. They put the case that previous approaches, in their assessment, lack the essential capacity to incorporate complex terrain and "the essential dynamics and physics" of tropical cyclone (TC) behaviour where "the accuracy of wind speeds over urban (sic) is of critical importance". The use of a diagnostic 3D numerical boundary layer model is central to their thesis.

While the desirability of such an approach can be supported, where practical, it is ironic that the demonstrated model skill is so poor at reproducing recent historical TC winds in areas where there is a significant amount of quality data available and in mostly flat landscapes. In no way does the model "compare favourably" with the displayed data. An 8 to 10 m/s error band in any hindcast event intended to assist, for example, offshore engineering design or sensitive onshore high-rise design, would be regarded as completely unacceptable. For insurance losses where damage is noted by the authors to be additionally highly nonlinear with wind speed, it would be massively unreliable. The

stated need to apply an empirical "20% adjustment for urban areas" is not only completely inconsistent with the theoretical high ground being argued but is symptomatic of a modelling system that has some significant problems.

As you state below, the challenge of developing a globally applicable approach is that the accuracy at the individual event level will be poorer than for a model developed for individual events or specific regions. This means that for our model evaluation for the subset of observation-rich storms over the U.S. we do not expect that our model should be able to compete with individual event-level approaches that assimilate far more observational data. A globally applicable approach has a number of unique benefits that derive from its small amount of required input data and physical response of the boundary layer winds to terrain. These benefits include generating events in data sparse-regions, generating synthetic events, and application to downscaling TC tracks from global climate models. The case for a global approach will be made stronger in the revised manuscript.

We note that recently published work that used similar quantities of input data and local wind multiplication factors to account for terrain features (Tan and Fang, 2018) also showed typical errors of 8 to 10m/s. Their Fig. 6 shows comparison with observations for 36 TCs during 1970–2014 for 25 stations in Hainan Island, China. They show 8 to 10m/s errors are typical in the 10-minute sustained winds. Our approach therefore compares favorably with another global modeling approach.

In addition, we also note this magnitude of error are also present in hurricane surface wind vectors utilizing C-band dual-polarization synthetic aperture radar observations, when compared to collocated QuikSCAT-measured wind speeds. For the case of Hurricane Bill Zhang et al. (2014) (Their Fig. 9a and 9b, copied here below) shows scatter (over the ocean) similar in magnitude to our scatter.

Fig. 9. (a) SAR-retrieved wind speeds from the C-2POD model vs QuikSCAT-measured wind speeds, (b) SAR-retrieved wind speeds from the CMOD5.N model vs from the C-2POD model, (c) SAR-retrieved wind directions vs QuikSCAT-measured wind directions, and (d) vector correlation of wind directions from SAR and QuikSCAT (sample size is 4). Hurricane Bill winds from SAR and QuikSCAT are acquired at 2226 and 2254 UTC 22 Aug 2009, respectively.

The revised manuscript will tone down the assertion that our model compares favorably with observations and will state that it compares favorably with recently published work.

Tan, C. and Fang, W.: Mapping the wind hazard of global tropical cyclones with parametric wind field models by considering the effects of local factors. International Journal of Disaster Risk Science, 9(1), 86-99, https://doi.org/10.1007/s13753-018-0161-1, 2018.

Zhang, B., Perrie, W., Zhang, J.A., Uhlhorn, E.W. and He, Y., 2014. High-resolution hurricane vector winds from C-band dual-polarization SAR observations. Journal of Atmospheric and Oceanic Technology, 31(2), pp.272-286.

We agree that our use of a 20% correction factor for winds over urban areas is weak. On further consideration, and in response to similar concerns from other reviewers, we decided to remove this bias correction factor from this study. The original factor of 20% was determined by comparing our simulations with surface station data in urban areas for a subset of 8 landfalling

U.S. hurricanes. This bias correction step was added to aid application of the dataset but clearly patches over an underlying problem.

Holmes (2007) found that the roughness length for urban areas can vary between 0.1 and 0.5m for suburban regions and rise to between 1 and 5m for densely packed high-rises in urban centers. Our model uses a single roughness length for all urban areas (suburban and city centers) of 0.8m and this was taken from the MODIS land use dataset – the same as used in the Weather Research and Forecasting (WRF) model. This value is too high for suburban areas, where a value closer to 0.2 is typical (Yang et al. 2014). Depending on the specific siting of the wind observing stations, it's probable that the introduction of multiple urban categories with different roughness lengths would improve our low wind speed bias. This detailed investigation is beyond the scope of this paper and we choose to leave this for future work.

Holmes, J. D., 2007. Wind loading of structures. 2nd ed. London and New York, Taylor & Francis

Yang, T., Cechet, R.P. and Nadimpalli, K., 2014. *Local wind assessment in Australia: Computation methodology for wind multipliers*. Geoscience Australia. 2014/33.

The Challenge of a Global Approach and the Expected Benefits

The problem with developing tools for global application unfortunately means that accuracy is inevitably impacted by the need to adopt spatially and/or temporally compromised globally available datasets. This situation limits, and actively dissuades, examination of the myriad of fine scale site-specific influences on the TC surface wind during a specific event that cannot be ignored. These have traditionally been transparently handled by reference to standard exposure and application of statistically based boundary layer turbulence approaches. The authors' more complex and computationally demanding approach needs to demonstrate at least a comparable utility.

Please see our comment above. Our approach shows comparable utility to a recently published globally applicable approach that combines parametric wind profile modeling with local wind multiplication factors.

In any case, it is not clear what practical application there is in producing such a global (deterministic) event set, given that the essential need is for risk management that implicitly requires a probabilistic approach. Cherry-picking of historical events does not yield firm statistical guidance and any such results will most likely be less reliable than any regional wind speed risk assessment that is based on (even sparse) long term data sets, given that aggregation of sites is often justified. In spite of the practical challenges in this space, the various engineering design standards around the world have overtime assembled realistic and likely suitably conservative wind risk frameworks, all of which embody the need to allow for height, terrain and topography effects.

A deterministic event set is of significance importance to the risk management industry for two reasons. Firstly, it allows for the validation of the wind field module of tropical cyclone catastrophe models (probabilistic approach). This module is a critical component of these models, and they do not use a physical model for representing the wind speed and rely on site coefficients to represent friction and topography. The module is used to generate stochastic wind fields and historical wind fields for a selection of events. A global deterministic event set based on a

physical model is therefore an excellent tool for validating catastrophe model's wind module via a comparison of historical wind footprints. Secondly, modelling as-if losses from historical storms is a widely used approach to stress test reinsurance structures to ensure companies have adequate protection, and can also be used in submissions to regulators. A global event set is an excellent basis for building scenarios to test as-if losses. These points will be emphasized in the revised manuscript.

The model's development is touted as valuable for insurance-related purposes. However, the principal cause of increasing world-wide losses for insurers is, together with uninformed risk-based planning, the failure to implement known good design, construction and inspection practices for residential development. The importance of globally modelling large scale terrain influences is also overstated given that, compared with typically nearly-flat or undulating conurbations, there is negligible insurance exposure to wind hazard in areas of very high or steep terrain.

We fully agree that one of the principal causes of increasing losses is poor implementation of known good construction practices. Indeed, Simmons et al. (2018) quantified the importance of a strong and well enforced wind building code for reducing insured wind losses in Florida.

There is an industry need for ever more accurate modelling of risk, often for single sites. Representing terrain influences is well known to influence wind speed, and mountainous countries often have both a high risk of tropical cyclones and significant insurance exposure, for example the Philippines and Japan. Therefore we believe an accurate representation of the influence terrain is very important, it is also included in all recent respected proprietary catastrophe models. This point will be made in the revised manuscript.

Simmons, K.M., Czajkowski, J. and Done, J.M., 2018. Economic effectiveness of implementing a statewide building code: the case of Florida. Land Economics, 94(2), pp.155-174.

**Comments on Method**

(2.) The step that "removes an estimate of the asymmetry due to storm motion" from the surface wind relies on the assumption that historical Vmax do reliably include such an influence. While Dvorak, for example, implies that is the case there is no specific allowance in the methodology. Hence the adopted empirical adjustment likely has little merit in terms of overall accuracy.

This is an important point. We agree that the extent to which asymmetry due to forward speed is included in best track estimates of Vmax may be questionable. But given that these winds are Earth relative measurements we assume that the benefits of removing an uncertain estimate of asymmetry outweighs the cons of not doing so. In this assumption, we follow the approach of others (e.g., Chavas et al. 2017). This point is made in the revised manuscript.

Chavas, D. R., Reed, K. A. and Knaff, J. A.: Physical understanding of the tropical cyclone windpressure relationship. Nature communications, 8(1), p.1360, https://doi.org/10.1038/s41467-017-01546-9, 2017.

In quoting Harper, Kepert and Ginger (2010) (aka the WMO wind averaging guidelines - hereafter HKG), the assumption that numerically modelled winds calculated at a small timestep are representative of so-called 1-min sustained winds is incorrect. Section 1.6 of HKG specifically advises on that topic noting that numerical models without explicit

eddy representation only estimate mean wind speeds, not gusts such as the so-called 1-min sustained wind. However, correcting for that (e.g. per HKG Table 1.2) will likely have no specific effect on the model performance.

Our numerically modeled winds are calculated on a timestep of a few seconds. This means that the model can only resolve wind variations of about 4 to 7 times the model timestep (depending on the variability of the flow). The instantaneous model outputs are therefore by no means the instantaneous wind. Rather, they are closer to the 1-minute average wind. It is true that our numerical modeling does not explicitly resolve turbulence. It parametrizes eddies. We agree with section 1.6 of HKG that the outputs of numerical models without explicit turbulence should be considered as the mean wind. And we consider the 1-minute wind to be a mean wind speed and not a gust measure.

(2.1) The authors state that the model is "agnostic to the source of the track data" as though that is some advantage, whereas the vast majority of historical datasets consist only of (lat, lon, Vmax) estimates with acknowledged high variability between agencies and also over-time. Rmax is also noted to be an essential parameter but is only recently available in some regions and not transparently derived. While these drawbacks are unavoidable, it further emphasises the challenge of using any (global) historical track data without critical assessment and expecting a high level of accuracy in the estimation of terrain-sensitive surface winds.

The cyclone-scale footprint will only be as good as the input best track data and the wind profile model. We also agree that details of how parameters such as Rmax are derived in historical track datasets are not transparent and are likely to vary wildly through time and between agencies. We will remove the statement that the model is "agnostic to the source of the track data" because while the model will function using different data sources, the accuracy of the cyclone-scale footprint will be very sensitive. This point will be made in the revised manuscript.

(2.2) While the Willoughby profile may be superior to some others for a hands-off global application it still requires an outer scale assumption and to note that TC scale, and its temporal evolution, is a critically important parameter in accurately modelling surface winds. The adopted land use surface roughness, with a scale of about 1 km, seems reasonable enough but should also be verified by example for the set of modelled storms. To note also that the references cited for drag coefficients are very dated and the authors could adopt more recent evidence to better suit their argued approach.

In response to this comment and a similar comment from another reviewer, the revised manuscript will better highlight the assumptions needed to model the outer winds of the Willoughby profile. Firstly, we now detail that the length scale for the transition region across the eyewall is set to 25km when Rmax is greater than 20km and set to 15km otherwise. For the shape of the vortex outside Rmax, we hold the faster decay length scale fixed at 25km, following the recommendation of Willoughby et al. (2006), and allow the second length scale (*X1*) and the contribution of the fast decay rate (*A*) to vary the shape of the profile. For the slower decay length scale, we now include note that Willoughby et al. (2006) show a wide range of the second length scale occurs in nature, with their Fig. 11 showing it can range from about 100 km to over 450 km. We therefore choose to allow *A* and *X1* to vary with the readily available parameters Rmax, Vmax and latitude, following Eqn. 11 in Willoughby et al. (2006).

The effects of the adopted land use surface roughness is included in the evaluation of the subset of U.S. storms shown in Fig. 4.

We will include an updated reference for the variation in surface drag over the ocean with wind speed. And we choose to keep the key early reference to Garratt (1977).

Comments on Results and Evaluation

(3.) The Puerto Rico example illustrates the intention and ability of the model to introduce terrain and topographic variability into its results but, without any verification, is otherwise meaningless and simply an "artist's impression".

We agree with the need to compare with available observations. This Puerto Rico example will be expanded (also in response to comments from other reviewers) to better demonstrate the effects of adding KW01, surface roughness, and topography. This expanded analysis will include comparison with the available surface station data.

(4.) To note that the use of 3-h sampled wind data is typically inappropriate for TC passages within, say, 100 km and may be a principal source of the poor comparisons. Application of HKG Table 1.1 is also dubious given the 3-h sampling but more so because of its limited and nominal exposure classes, which appear inconsistent with the aim of deriving fine scale surface winds. HKG Section 1.2 says "The aim has been to provide a broad-brush guidance that will be most useful to the forecast environment rather than a detailed analytical methodology" and "In particular, post analysis of TC events should seek to use the highest possible site-specific analytical accuracy for estimating local wind speeds. This would include consideration of local surface roughness, exposure and topographic effects when undertaking quantitative assessments of storm impacts." This implies an approach like Powell et al. (1996) is needed in such cases. Again, these oversights in applying HKG will likely have little effect on model performance, but do point to a lack of rigour in matters of wind magnitude adjustment.

Thank you for this comment. We apologize that made a mistake in the text. In the original manuscript we incorrectly stated that "points are shown for all instances of the observed time falling within 3 hours of the model time." This didn't make sense, and will be corrected to "comparisons between observations and model are made using model time within 5 minutes of the observation time". We were able to do this because we output the model fields every 10 minutes.

We agree that the application of HKG Table 1.1 to scale the observations to the 1-minute wind is limited by the nominal exposure classes. But we also don't have the resources to conduct a detailed investigation into the specific exposure of each surface observing station. We therefore choose to leave this as an uncertainty in the evaluation. Over many surface observing sites over many landfalling events this uncertainty should approach zero if the effect is random. Though we agree that there is a clear need to conduct detailed site-specific comparisons in future work. This point will be made in the revised manuscript.

As noted previously, the demonstrated performance of the model is very poor compared with the numerous less-complex examples that are cited. In spite of the authors' tendency to downgrade the utility of H\*WIND I would instead encourage pursuing such comparisons in order to locate the model deficiencies and improve its performance for the demonstration storms.

In our previous response we made the case that our approach is at least comparable to a recently published globally applicable model. It was not our intention to downgrade the utility of H\*WIND. H\*WIND has many benefits over our approach such as including the effects of storm-scale asymmetry unrelated to forward speed. We will rephrase the references to H\*WIND accordingly. Perhaps the only major disadvantage of H\*WIND is that data are only available for a subset of U.S. storms.

One challenge of evaluating our model using H\*WIND is the need for assumptions for how to convert our model wind speeds to the open terrain representation in H\*WIND. While a comparison against H\*WIND would be useful to explore the scales of variability that are not included in our modeling approach, we choose to restrict our evaluation to surface station data.

Comments on Global Landfalling TC Footprints

(5) The detailed commentary and interpretation of aspects of modelled landfall and inland wind decay characteristics in various localities seems to overlook the fact that the model is only reacting to the imposed "best track" intensity variation and therefore can have no better skill than offered by a simpler parametric approach. Surely fully dynamic modelling (e.g. HWRF or similar) is needed to reliably explore such impacts.

Our modeling approach only allows for one-way forcing by the gradient wind and pressure profiles on the tropical cyclone boundary layer. Possible upscale effects of local terrain back on the entire TC structure is not permitted and is therefore missed. For example, studies have shown that the TC track itself can be impacted by high mountain ranges (see the discussion in Ramsay and Leslie 2008). Our approach contains these effects to the extent they are contained in the input best track data. It is correct that our gradient winds are reacting to the imposed best track, but the winds at 10m above the ground are reacting to the local terrain. We will state this separation of effects more clearly in the revised manuscript.

We agree that NWP models would capture more of the terrain effects and therefore would be better suited to study terrain effects on the inland wind decay. But one of the goals of our study is to explore the effects across a global dataset, and generating this dataset would be computationally impractical for 2km resolution NWP modeling.

---

## Author Comment (AC2) · 4 Nov 2019

Thank you for your useful comments. Our responses to your comments and the comments of the other two reviewers will greatly improve the manuscript. Below are our responses (in blue) to each comment in turn.

While responding to comments from all reviewers we found a bug in our code in the way we treat asymmetry. As described in the original manuscript, we first remove an estimate of the asymmetry due to forward speed from the input best track  $V_{max}$ . The portion removed is a function of the TC translation speed,  $V_a=1.173V_1^{-0.63}$ , following Chavas et al., (2017). We then add back an estimate of the asymmetry to the spatial 10m wind field diagnosed by KW01, again following Chavas et al., (2017). And in addition, we apply a factor that varies with radial distance from the storm center (the factor is equal to 1 at the radius of maximum winds and then decays with increasing radius) following Jakobsen and Madsen (2004). The bug was that the code missed adding back the  $V_c$ -dependent factor and only added back the radially-dependent factor. This caused us to add back the full value of  $V_c$  at  $R_{max}$  which caused too strong asymmetry, particularly for fast moving storms over Japan for example. In response, original figures 2, 3 and 4 will be corrected. The analysis in original figure 7 is for along-track winds which will not be affected by this bug fix.

Chavas, D. R., Reed, K. A. and Knaff, J. A.: Physical understanding of the tropical cyclone windpressure relationship. Nature communications, 8(1), p.1360, https://doi.org/10.1038/s41467-017-01546-9, 2017.

Jakobsen, F. and H. Madsen, H.: Comparison and further development of parametric tropical cyclone models for storm surge modeling. Journal of Wind Engineering, 92, 375-391, https://doi.org/10.1016/j.jweia.2004.01.003, 2004.

The manuscript proposed a new modeling system for generating tropical cyclone (TC) wind, which consists of a parametric radial profile model, the non-linear boundary layer model (KW01), and the terrain effects. The authors presented a case of hurricane Maria and Wilma and verified the model using landfalling storms. Then, the authors discussed the impact of terrain on the changes of TC winds over the South East US, Taiwan and Eastern China, and Eastern Australia. Overall, I like this approach and can think of many applications of this model. One criticism I have is that the advantage of using KW01 was not shown or discussed with evidence. In Section 2.3, The authors described KW01 as the key advancing component of the modeling system. At many other places, the author says that their system contains more dynamical processes than other existing tools. However, none of the results shown here can isolate the positive impact of using KW01. I suggest conducting additional simulations using Willoughby's wind with an empirical factor applied to get winds at10m height, plus the terrain effect. By comparing these new simulations with the Willoughby+KW01+terrain, we can see the advantage of (or differences caused by) KW01.

The suggestion to better demonstrate the advantage of using KW01 is a good one. We agree that this is needed, given that we state that the use of KW01 is the main advance of our modeling approach. In the revised manuscript we will extend the analysis of Hurricane Maria over Puerto Rico to better demonstrate the advantage of KW01, and its interactions with surface roughness and topography.

The original analysis (shown in the original Fig. 2) compared Willoughby, Willoughby+KW01, and Willoughby+KW01+terrain. The new Fig. 2 will include two new simulations:

- 1) A simulation using Willoughby+KW01+surface roughness without topography. This simulation will isolate the effect of surface roughness from the effect of topography.
- 2) A simulation using Willoughby+KW01+ topography but with surface roughness equivalent to that of the ocean over the entire domain. This simulation will isolate the effect of topography from the effect of surface roughness. We will compare our topographic effect with topographic multiplication factors used in Yang et al. (2014).

We will include difference plots to show the impact of each modeling component in turn.

In addition, we will add a comparison of each simulation to the available surface station observations to get a sense of the added value of each modeling component. We will also include a figure showing a snapshot of the model output wind field as Maria makes landfall, to show any evidence for strong deviation of winds around local hills.

Regarding adding a simulation that applies an empirical factor to Willoughby, we explored the possibility of calculating wind multiplication factors that account for terrain and topographic effects. A recent summary of wind multiplication factors used by international and national wind engineering codes found large differences in the level of complexity in the approaches and assumptions (Yang et al. 2014). As described in Tan and Fang (2018), calculating the topographic multiplication factor is non-trivial and requires a number of assumptions about the overlapping influence of nearby peaks within complex terrain and how this varies with wind direction. The resulting wind field would then be very sensitive to our assumptions. Developing expertise in constructing topographic multiplication factors is beyond the scope of this study, and so we choose to respond to this point by adding a description of the differences in the wind multiplication factor approach and our approach.

In the absence of inland observations an alternative approach to model evaluation is to compare with a simulation using a full atmosphere numerical weather prediction model. We were able to acquire data from a Weather Research Forecasting model simulation of Maria. Our intention was to use this simulation to compare the terrain response of KW01 with the NWP model simulation. However, we found that WRF has a problem with dropping the 10-m winds too much over land and immediately at the coast. Fixing this problem was beyond the scope of this paper.

Tan, C. and Fang, W.: Mapping the wind hazard of global tropical cyclones with parametric wind field models by considering the effects of local factors. International Journal of Disaster Risk Science, 9(1), 86-99, https://doi.org/10.1007/s13753-018-0161-1, 2018.

Yang, T., Cechet, R.P. and Nadimpalli, K., 2014. *Local wind assessment in Australia: Computation methodology for wind multipliers*. Geoscience Australia. 2014/33.

Below are a few minor comments and questions 1. Page 4, line 2. While I understand the advantage of using KW01 instead of a simple empirical model, this sentence sounds vague. Please elaborate more on what the additional dynamical effects are.

**We will include a brief discussion of the additional dynamical effects.**

2. The first paragraph of section 5 (page 11) should belong to Section 2.1.

This paragraph will be moved as suggested.

3. Page 5, Line 15: Please mention the TC boundary layer height used in this study as well. Is the model performance sensitive to the TCBL height?

Page 7, Line 8 states that the model top is held fixed for all simulations at 2km. This was chosen to be above the typical height of super-gradient jets. We also state that 'While the boundary layer height likely varies substantially across global TCs, we choose to keep this fixed in the absence of readily available data'. We also note that the TCBL height varies strongly with radial distance from the center of a given storm (Kepert et al. 2012). The height also depends on the specific definition of TCBL top. We have not explored sensitivity to our model top, but agree that it is a key parameter that will affect other aspects of model setup such as the factor used to inflate the input best track Vmax from the surface to model top. This unexplored model sensitivity will be acknowledged and discussed in the conclusions.

Kepert, J. D.: Choosing a boundary layer parameterization for tropical cyclone modelling. Monthly Weather Review 140, 1427–1445, https://doi.org/10.1175/MWR-D-11-00217.1, 2012.

4. Page 7, L11 'running for 24 hours for each forcing update is computationally impractical.' I am surprised to see that running 24 hours of KW01 is computationally impractical. What is the computational cost of KW01, and how is it compare to the computational cost of 2-km WRF.

I am asking this is because, for assessing wind risk, the most significant advantage of a simplified wind generator v.s. a full-physical model is its low computational cost. If running KW01 is computationally expensive, this system will not be able to use for real risk assessment, which (I thought) is one (and probably the most important one) of motivations of this work. (The other motivation is to understand wind risk over complex terrain using historical cases. For this purpose, we can always run WRF or other mesoscale models which may generate more realistic winds than KW01)

Thank you for raising this important point. The simulation wall-clock time strongly depends on the number of grid points. For the large domains needed to capture the long tracks of fast-moving storms (over Japan or the Northeast U.S., for example) the wall-clock time is substantially longer than using wind profile models alone. The most expensive domain, over Japan, is 900 X 1100 X 18 grid points using 2-km grid spacing and a 2 second timestep. A 24-hour simulation took 6 hours wall-clock time on 36 cores. Smaller domains run at 4km run much quicker.

Running a 24-hour period for a single event is therefore computationally quite practical. But running for 24 hours for each 10-minute forcing update to ensure full equilibrium is reached for each forcing update would rapidly increase computational demands to impractical levels.

Running the WRF model over the same domain would cost more due to the higher number of vertical levels, and a greater number of physical processes. We have not run WRF over these specific domains so we are not able to provide a computational cost. Even if it were feasible to run WRF for all 714 historical cases simulated here, future applications of our modeling approach to large numbers of synthetic TC tracks would presumably become impractical for WRF.

We will include a short discussion of the computational cost of our modeling approach at the end of the conclusions section.

5. Page 9, L24. Do you mean the maximum wind speed recorded at the station during the lifetime of the storm, which is different from the storm lifetime maximum wind speed (which is usually one value per storm)?

We don't see this specifically mentioned on page 9, L24. But your point is correct. We do indeed mean the max wind speed recorded at the station during the lifetime of the storm. This point will be corrected throughout the manuscript.

6. Page 11, L1: Where is this 20% bias correction factor comping from? Is it universally applied to all simulations?

On further consideration, and in response to similar concerns from other reviewers, we decided to remove this bias correction factor from this study. The original factor of 20% was determined by comparing our simulations with surface station data in urban areas for a subset of 8 landfalling U.S. hurricanes. This bias correction step was added to aid application of the dataset but clearly patches over an underlying problem.

Holmes (2007) found that the roughness length for urban areas can vary between 0.1 and 0.5m for suburban regions and rise to between 1 and 5m for densely packed high-rises in urban centers. Our model uses a single roughness length for all urban areas (suburban and city centers) of 0.8m and this was taken from the MODIS land use dataset – the same as used in the Weather Research and Forecasting (WRF) model. This value is too high for suburban areas, where a value closer to 0.2 is typical (Yang et al. 2014). Depending on the specific siting of the wind observing stations, it's probable that the introduction of multiple urban categories with different roughness lengths would improve our low wind speed bias. This detailed investigation is beyond the scope of this paper and we choose to leave this for future work.

Holmes, J. D., 2007. Wind loading of structures. 2nd ed. London and New York, Taylor & Francis

Yang, T., Cechet, R.P. and Nadimpalli, K., 2014. *Local wind assessment in Australia: Computation methodology for wind multipliers.* Geoscience Australia. 2014/33.

7. Page 11. L12-14 belongs to the figure caption of Fig. 6, not in the main text.

This text will be moved to the figure caption of Fig. 5 (the figure that shows all domains).

8. Figure 7 and the related discussion. Did you check the enhanced vertical diffusion and vertical advection in KW01? Can you show some analysis of these enhanced features? There is a lag between the terrain and the wind gradient. Why?

Figure 7 shows how the inland gradient of wind speed varies with distance inland from the coast, together with the terrain height. The gradient of wind speed and the terrain height is the along-track average over all simulated storms by region. The gradient of the wind speed is also the net effect of not just topography but also variations in surface roughness and the overall inland decay according to the input best track Vmax. This makes it challenging to isolate the processes driving the inland wind speed gradient. We will highlight this complexity in the revised manuscript, and tone down our asserted mechanism to a suggested mechanism.

Carefully constructed idealized experiments would be needed to isolate the processes (enhanced vertical diffusion and vertical advection in KW01) driving wind acceleration on the upwind slopes and crests of terrain features. We choose to leave this investigation for another study that would focus more on the process-level understanding rather than a global assessment as presented here. This point will be discussed in the conclusions.

The distance-rate-of-change in wind speed shows increases (or for some regions, a lessening of the inland decay) along the upwind slopes up to the crest (where, for example in Fig. 7b the wind gradient switches from positive to negative). The lee sides show some evidence of accelerated inland wind decay. This is similar to the topographic effect on the winds shown for the single storm Maria in Fig. 2. We don't see a strong lead-lag effect.

9. Willoughby et al. 2006 is missing in the references.

Thank you for spotting this oversight. It will be added.

---

## Author Comment (AC3) · 5 Nov 2019

Thank you for these useful comments and list of references. Our responses to your comments and the comments of the other two reviewers will greatly improve the manuscript. Below are our responses (in blue) to each comment in turn.

While responding to comments from all reviewers we found a bug in our code in the way we treat asymmetry. As described in the original manuscript, we first remove an estimate of the asymmetry due to forward speed from the input best track $V_{max}$. The portion removed is a function of the TC translation speed, $V_a = 1.173 V_t^{0.63}$, following Chavas et al., (2017). We then add back an estimate of the asymmetry to the spatial 10m wind field diagnosed by KW01, again following Chavas et al., (2017). And in addition, we apply a factor that varies with radial distance from the storm center (the factor is equal to 1 at the radius of maximum winds and then decays with increasing radius) following Jakobsen and Madsen (2004). The bug was that the code missed adding back the $V_t$-dependent factor and only added back the radially-dependent factor. This caused us to add back the full value of $V_t$ at $R_{max}$ which caused too strong asymmetry, particularly for fast moving storms, over Japan for example. In response, original figures 2, 3 and 4 will be corrected. The analysis in original figure 7 is for along-track winds which will not be affected by this bug fix.

Chavas, D. R., Reed, K. A. and Knaff, J. A.: Physical understanding of the tropical cyclone wind-pressure relationship. Nature communications, 8(1), p.1360, https://doi.org/10.1038/s41467-017-01546-9, 2017.

Jakobsen, F. and H. Madsen, H.: Comparison and further development of parametric tropical cyclone models for storm surge modeling. Journal of Wind Engineering, 92, 375-391, https://doi.org/10.1016/j.jweia.2004.01.003, 2004.

**Review of "Modelling Global Tropical Cyclone Wind Footprints" by James M. Done et al.**

**Summary**

The MS describes a method for automated modelling of tropical cyclone winds, both instantaneous fields and the maximum wind swath over the life of the storm. An axisymmetric representation of the gradient-level wind is derived using inputs from, for example, a best track database. Then a nonlinear boundary-layer model is used to calculate the winds throughout the boundary layer, including at the surface (10 m), from this gradient-level wind, accounting for storm motion, heterogeneous surface roughness and topography.

There are three potentially serious flaws with this approach, relating (i) to the way the parametric profile of Willoughby et al (2006) has been used, (ii) to the likely inability of the nonlinear tropical cyclone boundary layer model to correctly model mountain waves, and (iii) to the authors' misapplication of the work of Harper et al (2010) in adjusting observed winds for different averaging periods. These are expanded upon below. In addition, some minor points where clarification is needed are noted.

**Use of the Willoughby et al. (2006) parametric profile**

The criticism here is not that the authors have chosen this profile – indeed, I consider it to be the most suitable tropical cyclone parametric profile presently available, because of its superior ability to fit observations. Rather, it is criticism of the way they have used it. The authors note that in section 2.2 that the "Holland et al. (2010) profile has the advantage of tying down the radial decay profile using an observation of an outer wind, say the radius of 34 knot winds" and go on to note that such observations are not always available. They then note that the Willoughby et al. (2006) profile has two exponential decay scales for the outer part of the vortex. In addition, the user must assign the relative weight of these two profiles, so there are three free parameters that determine the shape of the vortex outside of the radius of maximum winds (RMW), although in practice Willoughby et al. (2006) recommends that one length scale be held fixed at 25 km. Choosing values for the remaining two free parameters requires additional data; Kepert (2006a,b) and Schwendike et al. (2008)

describe the use of aircraft reconnaissance data for this purpose and discuss the associated difficulties. This choice can lead to substantial differences in the shape of the wind profile and hence the radius of gales, and Willoughby et al. (2006) show that a wide range of the second length scale occurs in nature, with their Fig 11 showing it can range from about 100 km to over 450 km. The authors of the MS under review not only omit to describe how they have chosen these crucial parameters; they also incorrectly assert that the Willoughby et al (2006) profile has "fewer required data inputs".

Thank you for this informed comment on our use of the Willoughby et al. (2006) parametric profile. We will remove the assertion that the Willoughby et al. (2006) profile requires fewer data inputs than the Holland et al. (2010) profile.

We agree that the outer radius of damaging winds can be highly sensitive to the choice of the free parameters. We did not provide sufficient detail on how the three free parameters are chosen. The revised manuscript will explain our choices.

Firstly, we will state that the length scale for the transition region across the eyewall is set to 25km when Rmax is greater than 20km and is set to 15km otherwise.  For the shape of the vortex outside Rmax, we hold the faster decay length scale fixed at 25km, following the recommendation of Willoughby et al. (2006), and allow the second length scale (*X1*) and the contribution of the fast decay rate (*A*) to vary the shape of the profile. For the slower decay length scale, we will note that Willoughby et al. (2006) showed that a wide range of the second length scale occurs in nature, with their Fig. 11 showing it can range from about 100 km to over 450 km. We will also note the difficulties in using aircraft reconnaissance data for this purpose (as in Kepert 2006a,b and Schwendike et al. 2008). Given that aircraft data are not uniformly available globally, and our intention to only use readily available data, we decide not to include these additional data sources for subsets of global historical events. Willoughby et al. (2006) demonstrated dependence on Vmax, and latitude, with the more intense low latitude TCs being more sharply peaked. We also have Rmax readily available. We therefore choose to allow the remaining two free parameters (*A* and *X1*) to vary with readily available parameters Rmax, Vmax and latitude, following equation 11 in Willoughby et al. (2006). This globally consistent approach is needed to allow relative risk assessments across regions.

**Use of the Kepert and Wang (2001) nonlinear tropical cyclone boundary layer model**

This model, and others like it, are the most sophisticated diagnostic models of the tropical cyclone boundary-layer presently available. However, there are two important areas in which the authors have failed to establish that their use of the model is appropriate.

Firstly, the model as originally formulated was written in storm-following coordinates. This enabled the efficient simulation of moving storms, since a smaller domain could be used, and indeed Kepert and Wang (2001) remains one of the few theoretical papers on the tropical cyclone boundary layer to consider the effects of storm motion. The authors describe some modifications to the model, which they state are to allow for a time-varying gradient wind field, and for landfall. While their description is unclear, it appears that they may have changed to earth-relative coordinates, for they

state that "a translation vector is added to the horizontal advection terms in KW01". Unfortunately, the governing equations are omitted, so it is impossible to be sure. However, they do note that "the proportion of the translation vector added reduces close to the surface due to surface friction". This is certainly incorrect; the whole of the coordinate system must move with the same velocity! Perhaps it is an attempt to allow for friction in the environmental flow, which the model (in its original form) assumes is equal to the translation vector. However, whether in earth-relative or storm-relative coordinates the model should be able to spin up the boundary layer of any environmental flow, and if in storm-relative coordinates does not require the addition of a translation vector. Perhaps the authors' modification is correct, but until they give equations this cannot be established, and as outlined above, their description doesn't sound correct.

Thank you for this comment. We agree that our original description was unclear. We also apologize for including some old and incorrect description of the model setup. The few lines of incorrect description was a legacy of some earlier model development and testing. The revised manuscript will clearly and accurately describe the final modeling approach that we settled on.

The original model coordinates of KW01 were storm-relative. The storm in this original coordinate system therefore did not move by definition. To account for the effect of storm motion, the original model includes the pressure gradient of the environmental flow in the forcing pressure profile at the model top.

We changed the model to run in Earth-Relative coordinates. Each simulation is conducted on one of the 17 geographically-fixed regional domains shown in original Fig. 5. Our early model testing retained the effect of forward speed in the forcing of KW01 but we found KW01 strongly damped the asymmetry and we did not have the resources to find the cause. We therefore chose to take out the effect of forward speed from the forcing for KW01 and treat it in pre- and post-processing steps as depicted in the workflow shown in original Fig. 1. We also do not add the translation vector to the horizontal advection terms. That was a mistake in our description so thank you for pointing that out.

Our approach therefore misses any interaction effects between terrain and the asymmetrical component of the storm wind field. The importance of these effects are unknown and we leave their inclusion for a future iteration of our model.

I note also that, since the Kepert and Wang (2001) model incorporates the effects of translation, the postprocessing step of adding on a motion asymmetry shown in Figure 1 is at best unnecessary, but most likely also incorrect.

As noted in our response above, we removed the effects of translation from KW01. Adding on a motion asymmetry was therefore necessary.

The second issue concerns the possibility of mountain wave activity. The model does allow for surface topography, although this facility has not before been the subject of published papers to my knowledge. However, it is unlikely it would accurately represent mountain wave activity, because of the shallow depth of the domain – modelling studies of mountain waves typically consider at least the full depth of the troposphere. Although some of the discussion in the introduction could be interpreted as evidence against mountain waves, it is far from rigorous – for instance, the authors note that the Froude number will be high without considering that this will also depend on the flow geometry. In addition, they note the

"quasi-neutral stability"; while this is plausible near the eyewall provided one considers moist stability, it is incorrect at larger radius as shown by the observational composites of Zhang et al. (2011).

Thank you for this useful comment. We can see how our discussion of full NWP model simulations by Ramsay and Leslie (2008) may have been interpreted as an indication that mountain wave activity was represented in our modeling approach. Our model top at 2-km cuts off the free troposphere needed for large-scale mountain waves that can bring free-atmosphere winds down to the surface. It's also unlikely that our model has the resolution to capture flow-separation turbulence downwind of crests and escarpments.

The revised manuscript will more clearly articulate the physical terrain effects permitted by the model. These include convergence, vertical diffusion and vertical advection on windward slopes and crests resulting in locally strong low-level shear and TKE production. In addition, the vertical boundary layer structure allows the potential for super-gradient jets (Franklin et al. 2003; Kepert and Wang 2001) to influence winds in high terrain. Finally, the time dimension allows for upwind effects due to upwind terrain variations and terrain to be incorporated.

In the revised manuscript, our discussion of the Froude number will acknowledge the contribution of mountain geometry. For high mountains, the wind has a greater potential to become blocked. On closer inspection we didn't see any evidence of substantial blocking (upstream deceleration) in any of our runs. This will be discussed further for the case of Maria over Puerto Rico in an expanded Fig. 2. We will also acknowledge that our modeling approach doesn't directly allow any blocking to affect the TC track, although the observed tracks that are used do include such blocking effects.

We agree that a neutral boundary layer is not guaranteed at large radii. Kepert (2012) indicates increasing static stability as subsidence increases at these larger radii. Even so, measurements of turbulent fluxes in high-wind environments between outer rainbands by Zhang et al. (2009) find shear production and dissipation to be the dominant source and sink terms of TKE.

Zhang, J. A., W. M. Drennan, P. G. Black, and J. R. French, 2009: Turbulence structure of the hurricane boundary layer between the outer rainbands. J. Atmos. Sci., 66, 2455–2467

**Conversion of wind speed averaging periods**

The authors have adjusted surface wind observations for averaging period, claiming as justification the work of Harper et al. (2010). This is incorrect. Harper et al. (2010) emphasise that their conversion factors are to be used for tropical cyclone intensity, and that they should **not** be used for wind observations. Please refer to the third paragraph of the executive summary, section 1.3 and appendix E of that report.

We understand that the shorter averaging period can only be considered a gust in the context of a longer averaging period over which the wind is considered steady (i.e., constant variance/turbulence). Estimating the 1-minute 'gust' from buoy-observed 10-minute mean wind should therefore be appropriate. But estimating the 1-minute 'gust' from the surface station-observed 2-minute wind is questionable since the 2-minute wind is not necessarily a measure of the mean wind. This 2-minute wind could be higher or lower than the true mean wind.

However, we still desire to use these 2-minute observations for model evaluation. We therefore choose to use the factor to estimate the 1-minute wind from the 2-minute wind, despite the inconsistency with statistical theory. In this, we assume that errors arising from this factor application are i) similar in magnitude to the uncertainty in the definition of the 1-minute wind from numerical model data, and ii) similar to errors due to gusts deviating from statistical theory due to winds not in equilibrium with the variable surface conditions.

**Further comments**

Page 3 line 2, the Willoughby et al (2006) profile is not intended for surface winds.

In the revised manuscript we will more accurately state that the Holland et al., (2010) profile, for example, models the surface winds directly whereas the Willoughby et al., (2006) profile models the gradient-level winds and an extra step is needed to determine the surface winds.

Page 5 line 8, and elsewhere, the authors claim to use an "average value of Rmax around the storm". This seems strange, Rmax is usually not regarded as having significant asymmetries, unlike say R34.

This comment about taking an average value of Rmax around the storm will be removed.

Page 5 line 16, 500 m is too low here, as it is either at or below level of the supergradient jet.

We will change this to be consistent with the definition of boundary layer height based on the depth of the inflow. We now explain that 'One definition for the boundary layer height is the depth of the inflow, defined as the height where the radial inflow falls to 10% of the peak inflow. Using radiosonde ascents in 13 hurricanes Zhang et al., (2011) find this height to be approximately 850m at the radius of maximum wind rising to approximately 1300m at larger radii.'

Page 6 line 8, the value of the eyewall surface wind factor of 1/1.32 = 0.75 is high compared to observations (Franklin et al. 2003, Powell et al. 2009) and theory (Kepert and Wang 2001).

Franklin et al. (2003) observed an inner core wind maximum at about 500m that increases to about 1km for the outer winds. They found a logarithmic profile below the wind maximum and a reduction of winds above due to the warm core. The result is a 700-hPa-to-surface wind factor of about 0.9 in the inner core. Kepert and Wang (2001) theory also has a factor of 0.9 in the inner core, with the factor decreasing to 0.75 in the outer winds. Knaff et al. (2011) took the Franklin et al. (2003) factor of 0.9 in the inner core and additionally reduced it by a factor of 0.8 to go from marine-exposure winds to terrestrial-exposure winds, giving a net factor of 0.9*0.8=0.72.

Initial testing using a variable factor for offshore and onshore track caused enhanced winds just offshore in response to the higher factor for the first inland track point. This is because the outer winds were still responding to the low surface roughness while being driven by stronger gradient winds. We therefore choose to hold the factor fixed and appropriate for inland winds.
This will be discussed in the revised manuscript.

Knaff, John A, Mark DeMaria, Debra A Molenar, Charles R Sampson, and Matthew G Seybold. 2011. "An Automated, Objective, Multiple-Satellite-Platform Tropical Cyclone Surface Wind Analysis." Journal of Applied Meteorology and Climatology 50 (10): 2149–66.

Page 6 line 31, the Kepert and Wang (2001) model cannot resolve turbulence, since it uses a fixed pressure field. In this, it is unlike recent high-resolution simulations of tropical cyclones by Nolan et al. (2014) and Stern and Bryan (2018).

We agree that Kepert and Wang (2001) cannot resolve turbulence, but it does parameterize the effects of turbulence through its turbulence scheme. This will be more clearly stated.

Page 7 line 2, the model does represent buoyancy, although probably not particularly well since it ignores moist processes.

In the revised manuscript we will more correctly state that the model does not represent buoyancy *well*. But for most TCs, and for the inner core region of the strongest winds, we expect buoyancy effects to be small

Page 9 line 12, it is unclear how figure 3a shows this correction.

In response to the concerns of the other two reviewers we decided to remove this 20% correction factor from this study. The original factor of 20% was determined by comparing our simulations with surface station data in urban areas for a subset of 8 landfalling U.S. hurricanes. This bias correction step was added to aid application of the dataset but clearly patches over an underlying problem.

Holmes (2007) found that the roughness length for urban areas can vary between 0.1 and 0.5m for suburban regions and rise to between 1 and 5m for densely packed high-rises in urban centers. Our model uses a single roughness length for all urban areas (suburban and city centers) of 0.8m and this was taken from the MODIS land use dataset – the same as used in the Weather Research and Forecasting (WRF) model. This value is too high for suburban areas, where a value closer to 0.2 is typical (Yang et al. 2014). Depending on the specific siting of the wind observing stations, it's probable that the introduction of multiple urban categories with different roughness lengths would improve our low wind speed bias. This detailed investigation is beyond the scope of this paper and we choose to leave this for future work.

Holmes, J. D., 2007. Wind loading of structures. 2nd ed. London and New York, Taylor & Francis

Yang, T., Cechet, R.P. and Nadimpalli, K., 2014. *Local wind assessment in Australia: Computation methodology for wind multipliers*. Geoscience Australia. 2014/33.

Page 11 line 1 and in the conclusions, the claim that the model shows "no large bias" in urban areas seems optimistic. At a radius of 300 km, the bias is about -10 m/s. At this radius, this is likely well over half the observed wind speed, hardly negligible!

We agree that stating "no large bias" is not accurate. In response (and in response to another reviewer's comment) we will change this to state that the model compares well with a recently published global wind swath modeling approach (Tan and Fang, 2018). Their approach using similar quantities of input data and local wind multiplication factors to account for terrain features also showed typical errors of 8 to 10m/s. See their Fig. 6 that shows comparison with observations for 36 TCs during 1970–2014 for 25 stations in Hainan Island, China.

Tan, C. and Fang, W.: Mapping the wind hazard of global tropical cyclones with parametric wind field models by considering the effects of local factors. International Journal of Disaster Risk Science, 9(1), 86-99, https://doi.org/10.1007/s13753-018-0161-1, 2018.

The term "storm lifetime maximum wind" is generally used to refer to the wind swaths (e.g. page 13 line 8) but is ambiguous since it could also refer to the storm's peak intensity. In this part of the MS, these maximum winds will generally not occur at the storm centre, but they are analysed in terms of along-track distance. How is this calculated?

Thank you for raising this important distinction, that another reviewer also noted. We agree that although we used common definitions, they are rather imprecise. Our use of 'storm lifetime maximum wind' refers to the maximum wind speed recorded at a specific location (grid point or observing station location) throughout the lifetime of the storm. The revised manuscript will be specific whenever this term is used.

For Fig. 7 specifically, the wind data are extracted from the wind swath at the location of the TC track. This data is therefore the storm lifetime maximum wind speed at the specific locations along the TC track. Then we composited all TC tracks for a given region about their track points of landfall, and took an average over all tracks. This gives the region-average along-track wind swath vs. distance inland. This calculation will be made clearer in the revised manuscript.

**References**

Franklin, J. L., M. L. Black, and K. Valde, 2003: GPS dropwindsonde wind profiles in hurricanes and their operational implications. Wea. Forecasting, 18, 32–44.

Harper, B. A., J. D. Kepert, and J. D. Ginger, 2010: Guidelines for converting between various wind averaging periods in tropical cyclone conditions, WMO/TD-No. 1555. World Meteorological Organisation, 64 pp.

Kepert, J. D. and Y. Wang, 2001: The dynamics of boundary layer jets within the tropical cyclone core. Part II: Nonlinear enhancement. J. Atmos. Sci., 58, 2485–2501.

Kepert, J. D., 2006: Observed boundary–layer wind structure and balance in the hurricane core. Part I: Hurricane Georges. J. Atmos. Sci., 63, 2169–2193. doi:10.1175/JAS3745.1

Kepert, J. D., 2006: Observed boundary–layer wind structure and balance in the hurricane core. Part II: Hurricane Mitch. J. Atmos. Sci., 63, 2194–2211. doi:10.1175/JAS3746.1

Nolan, D. S., J. A. Zhang and E. W. Uhlhorn, 2014: On the limits of estimating the maximum wind speeds in hurricanes. Mon. Wea. Rev., 142, 2814 – 2837. doi:10.1175/MWR-D-13-00337.1

Powell, M. D., P. J. Vickery, and T. A. Reinhold, 2003: Reduced drag coefficient for high wind speeds in tropical cyclones. Nature, 422, 279–283.

Schwendike, J. and J. D. Kepert, 2008: The boundary–layer winds in Hurricanes Danielle (1998) and Isabel (2003). Mon. Wea. Rev., 136, 3168–3192.

Stern, D. P. and G. H. Bryan, 2018: Using Simulated Dropsondes to Understand Extreme Updrafts and Wind Speeds in Tropical Cyclones. Mon. Wea. Rev., 146, 3901 – 3925. doi:10.1175/MWR-D-18-0041.1

Willoughby, H. E., R. W. R. Darling, and M. E. Rahn, 2006: Parametric representation of the primary hurricane vortex. Part II: A new family of sectionally continuous profiles. Mon. Wea. Rev., 134, 1102–1120.

Zhang, J. A., R. F. Rogers, D. S. Nolan, J. Marks, and D. Frank, 2011: On the characteristic height scales of the hurricane boundary layer. Mon. Wea. Rev., 139, 2523–2535, doi:10.1175/MWR-D-10-05017.1.

---

## Author Response (AR2)

*Only one technical correction – can the color scales and size of text in figure 3 be improved?*

Thank you for this useful comment. We have improved figure 3g by changing the color scale to be a categorical color scale that reflects the roughness lengths of the different land use types. In addition, we list the land-use types together with their roughness lengths, using a larger font. We assume that the other components are figure 3 are acceptable because they use the same color scales as used in figure 2.

[revised manuscript text omitted]